# Identification of human gene research articles with wrongly identified nucleotide sequences

Yasunori Park[1], Rachael A West[1,2], Pranujan Pathmendra[1], Bertrand Favier[3], Thomas Stoeger[4,5,6], Amanda Capes-Davis[1,7], Guillaume Cabanac[8], Cyril Labbé[9], Jennifer A Byrne[1,10]

**Nucleotide sequence reagents underpin molecular techniques that have been applied across hundreds of thousands of publications. We have previously reported wrongly identified nucleotide sequence reagents in human research publications and described a semi-automated screening tool Seek & Blastn to fact-check their claimed status. We applied Seek & Blastn to screen >11,700 publications across five literature corpora, including all original publications in *Gene* from 2007 to 2018 and all original open-access publications in *Oncology Reports* from 2014 to 2018. After manually checking Seek & Blastn outputs for >3,400 human research articles, we identified 712 articles across 78 journals that described at least one wrongly identified nucleotide sequence. Verifying the claimed identities of >13,700 sequences highlighted 1,535 wrongly identified sequences, most of which were claimed targeting reagents for the analysis of 365 human protein-coding genes and 120 non-coding RNAs. The 712 problematic articles have received >17,000 citations, including citations by human clinical trials. Given our estimate that approximately one-quarter of problematic articles may misinform the future development of human therapies, urgent measures are required to address unreliable gene research articles.**

## Introduction

The promise of genomics to improve the health of cancer and other patients has resulted in billions of dollars of research investment which have been accompanied by expectations of similar quantum gains in health outcomes (1, 2). Since the first draft of the human genome was reported (3, 4), a series of increasingly rapid technological advances has permitted the routine sequencing of human genomes at scale (1, 2), and the increasing application of genomics to inform clinical care (1, 2, 5). Despite the now routine capacity to sequence the human genome, genomics research relies upon results produced by other research fields to translate genome sequencing results to patients (5, 6, 7). For example, although whole genome sequencing demonstrates that thousands of human genes are mutated or deregulated in human cancers (1), additional information is required to prioritise individual gene candidates for subsequent pre-clinical and translational studies (5, 6, 7).

A first step in triaging and prioritising gene candidates for further analysis is the consideration of available knowledge of predicted and/or demonstrated gene functions (5, 6, 7, 8). High-quality, reliable information about gene function is important to select promising gene candidates and to then progress these candidates through pre-clinical and translational research pipelines (8). This is supported by drug candidates with genetically supported targets being significantly more likely to progress through phased clinical trials (9, 10). However, in contrast to the sophisticated platforms that produce genomic or transcriptomic sequence data at scale, pre-clinical experiments typically analyse single or small numbers of genes through the application of more ubiquitous molecular techniques (6), some of which have been in routine experimental use for 15–30 yr. For example, gene knockdown approaches have been widely used to assess the consequences of reduced gene expression in model systems (6). Similarly, RT–PCR is frequently used to analyse the transcript levels of small groups of genes, either to confirm the effectiveness of gene knockdown experiments or in association with other experimental techniques. The widespread use and reporting of gene knockdown and PCR approaches reflect their low cost and accessibility, in terms of the necessary reagents, laboratory equipment and facilities, and the availability of technical expertise within the research community. As a consequence, the results of experiments using gene knockdown and/or PCR in the context of human research have been described in hundreds of thousands of publications that are retrievable through PubMed or Google Scholar.

[1]Faculty of Medicine and Health, The University of Sydney, Sydney, Australia   [2]Children's Cancer Research Unit, Kids Research, The Children's Hospital at Westmead, Westmead, Australia   [3]Université Grenoble Alpes, Translationnelle et Innovation en Médecine et Complexité, Grenoble, France   [4]Successful Clinical Response in Pneumonia Therapy Systems Biology Center, Northwestern University, Evanston, IL, USA   [5]Department of Chemical and Biological Engineering, Northwestern University, Evanston, IL, USA   [6]Center for Genetic Medicine, Northwestern University School of Medicine, Chicago, IL, USA   [7]CellBank Australia, Children's Medical Research Institute, Westmead, Australia   [8]Computer Science Department, Institut de Recherche en Informatique de Toulouse, Unité Mixte de Recherche 5505 Centre National de la Recherche Scientifique (CNRS), University of Toulouse, Toulouse, France   [9]Université Grenoble Alpes, CNRS, Grenoble INP, Laboratoire d'Informatique de Grenoble, Grenoble, France   [10]New South Wales Health Statewide Biobank, New South Wales Health Pathology, Camperdown, Australia

Correspondence: jennifer.byrne@health.nsw.gov.au

Experiments that analyse individual genes typically require nucleotide sequence reagents as either targeting and/or control reagents (8, 11). As nucleotide sequence identities cannot be deduced by eye, DNA or RNA reagent sequences must be paired with text descriptions of their genetic identities and experimental use (8, 11, 12, 13). The integrity of reported experiments therefore requires both the identities of nucleotide sequence reagents and their text descriptions to be correct (11, 12). Accurate reporting of nucleotide sequence reagents is also critical to permit reagent reuse across different experiments and publications (11). The ubiquitous description of nucleotide sequence reagents within the biomedical and genetics literature, combined with the routine pairing of nucleotide sequences and text identifiers, are likely to contribute to tacit assumptions that reported nucleotide sequence reagents are correctly identified. However, as nucleotide sequences cannot be understood by eye, we have proposed that nucleotide sequence reagents are susceptible to different types of errors (8, 11, 12). These error types represent the equivalent of spelling errors (12, 14, 15), as well as identity errors, where a correct sequence is replaced by a different and possibly genetically unrelated sequence (11, 12, 13, 16, 17, 18, 19, 20, 21).

The problem of wrongly identified nucleotide sequence reagents was recognised in the context of DNA microarrays in the early 2000s, where wrongly identified sequence probes affected the reliability and reproducibility of data from particular microarray platforms (22, 23). Our team subsequently identified frequent wrongly identified sh/siRNA's and RT–PCR primers in articles that commonly reported the effects of knocking down single human genes in cancer cell line models, where some articles also analysed common human genes across multiple articles and cancer types (12, 13). These articles were termed single gene knockdown (SGK) articles and were commonly authored by teams from China (13). SGK articles showed numerous unexpected similarities, such as similarities in textual and figure organisation, outlier levels of textual similarity, and the description of identical incorrect sequence reagents across articles that investigated different genes (11, 12, 13). We proposed that the similarities between and errors within SGK articles could reflect the undeclared involvement of organisations such as paper mills (13), which have been alleged to mass-produce fraudulent manuscripts for publication (24, 25). We proposed that the production of manuscripts at scale could underpin unusual degrees of similarity between publications, as well as features such as superficial explanations for the analysis of particular genes, and generic experimental approaches (8, 13, 24, 25, 26). We also proposed that producing many gene research manuscripts at minimal cost could involve writers with either an incomplete understanding of the experiments that they are describing and/or limited time for quality control (8, 24). These conditions could lead to wrongly identified nucleotide sequences being a feature of gene research articles from paper mills (8, 24) where incorrect sequences could also be reused across different manuscripts (11, 12, 13).

Our discovery of frequent wrongly identified nucleotide sequence reagents in human gene research articles led us to develop a semi-automated tool Seek & Blastn (S&B) to fact-check the reported identities of nucleotide sequence reagents in human research articles (12, 27). The S&B tool scans text to identify and extract nucleotide sequences and their associated text descriptors,

submits extracted nucleotide sequences to Blastn analysis (28) to predict their genetic identities and hence their targeting or non-targeting status, and then compares the predicted status of each nucleotide sequence reagent with the claimed status within the text (12). The Blastn results for each extracted nucleotide sequence are then reported and any sequence whose text identifier contradicts its Blastn-predicted targeting/non-targeting status is flagged as being potentially incorrect (12). Flagged nucleotide sequences are then subjected to manual verification, as described in our original publication (12) and an expanded online protocol (https://www.protocols.io/view/seek-amp-blastn-standard-operating-procedure-bjhpkj5n). Following descriptions of human gene research articles with wrongly identified nucleotide sequences and the S&B tool (12, 13), we aimed to apply S&B to different literature corpora to examine the frequency of wrongly identified nucleotide sequence reagents in different publication types. We also aimed to describe shared features of articles with wrongly identified sequences, such as country and institution of origin (13) and how problematic articles have been cited by subsequent research. We therefore used S&B to screen original research articles across five literature corpora, representing three targeted corpora and two journal corpora (Fig 1).

Targeted corpora were selected using specific keywords as literature search terms, which in some cases were combined with PubMed similarity searches of index articles. We had previously identified frequent wrongly identified nucleotide sequences in SGK articles that analysed the same human genes across multiple articles and cancer types (13). We therefore combined the human gene identifiers of 17 human genes, most of which had been analysed in previously reported articles (12, 13), with keywords used to identify SGK articles (13) to identify a targeted SGK corpus to be screened by S&B (Fig 1). As PubMed similarity searches using SGK articles identified articles that analysed the functions of different human miR's in cancer cell lines, we selected miR-145 to define a single miR corpus as a second targeted corpus (Fig 1). The miR-145 corpus was identified using keyword searches and PubMed similarity searches of two index articles from China, one of which described wrongly identified sequences. Finally, as we also noted examples of SGK and miR-145 articles that analysed the effects of drug treatments of cancer cell lines, we identified articles that described either cisplatin (29) or gemcitabine (30) treatment of human cancer cell lines and/or cancer patients as a third targeted corpus (Fig 1). The cisplatin and gemcitabine (C + G corpus) was identified using targeted keyword searches and PubMed similarity searches of seven index articles, including six articles from China that described wrongly identified sequences.

Although the keywords that were used to derive targeted corpora did not refer to author affiliations, articles in targeted corpora were selected using features of and/or index articles with incorrect nucleotide sequences (12, 13), where index articles were largely authored by hospital-based teams from China. We therefore complemented analyses of targeted corpora by screening all original and original open-access articles in *Gene* and *Oncology Reports* from 2007–2018 to 2014–2018, respectively (Fig 1). *Gene* and *Oncology Reports* were selected as representative examples of journals that had published articles with incorrect nucleotide sequences (12, 13), where *Gene* (published by Elsevier) encompasses

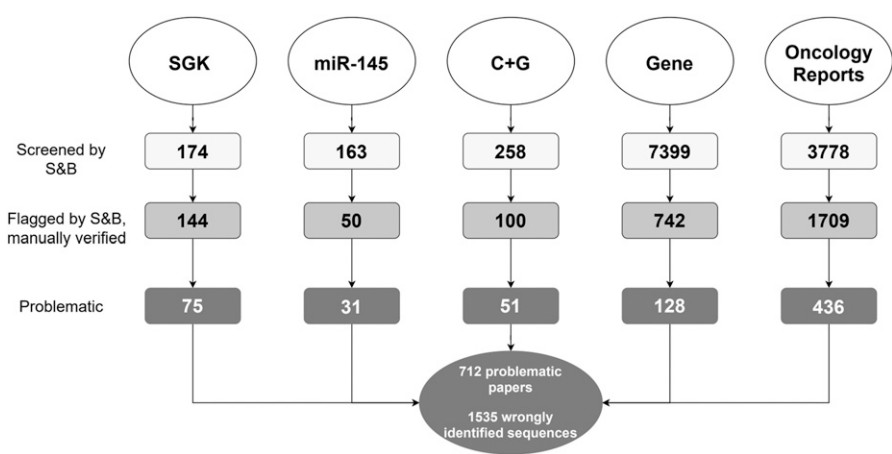

**Figure 1. Diagram describing the five literature corpora screened by S&B.**
For each corpus (top row), the diagram shows the numbers of articles that were (i) screened by S&B (white), (ii) flagged by S&B with sequences manually verified (grey), and (iii) found to be problematic by describing at least one wrongly identified nucleotide sequence (dark grey). Total numbers of problematic articles and wrongly identified sequences are indicated below the diagram, corrected for duplicate articles between the corpora.

a broad range of gene research across different species and *Oncology Reports* (published by Spandidos Publications) focusses on human cancer research.

Applying S&B to screen more than 11,700 articles across five literature corpora flagged 3,423 articles for manual verification, which identified 712 problematic articles with wrongly identified sequence reagents. Based upon our previous reports of articles with wrongly identified nucleotide sequences (12, 13), we analysed problematic articles according to their countries of origin and institutional affiliations. We then examined whether problematic articles have been cited in the literature and indexed by relevant gene knowledge bases, and how these articles could influence future clinical research.

## Results

### Analysis of targeted publication corpora

To extend our previous results from using S&B to screen gene research articles (11, 12), we used S&B to screen three targeted publication corpora (Fig 1) identified through literature searches that used specific keywords, and in some cases, PubMed similarity searches of index articles (12, 13). In all cases, the keywords that were used to derive targeted corpora did not refer to author affiliations, such as institution type or country of origin (see the Materials and Methods section).

### SGK corpus

We sought to identify all SGK articles for a set of 17 human genes (*ADAM8, ANXA1, EAG1, GPR137, ICT1, KLF8, MACC1, MYO6, NOB1, PP4R1, PP5, PPM1D, RPS15A, TCTN1, TPD52L2, USP39,* and *ZFX*), most of which had been analysed in previously reported articles (12, 13). Combining each of the 17 human gene identifiers with keywords previously used to identify SGK articles (13) identified 174 SGK articles published between 2006 and 2019 across 83 journals (Table 1). As most gene identifiers were selected from previously reported SGK articles (12, 13), the SGK corpus consisted of 41 (24%) previously

reported articles and 132 (76%) additional articles (Table 2). All 174 SGK articles analysed a single human cancer type (Table 2).

Across the 17 queried genes, we identified a median of eight articles/gene (range 3–20) that analysed a median of eight human cancer types (range 3–11) (Table 2 and Supplemental Data 1). Most (136/174, 78%) SGK articles named a single queried gene in their titles, with the remaining titles also referring to other human gene(s) and/or drugs, most frequently cisplatin (Supplemental Data 1). Most (159/174, 91%) SGK articles were published by authors from mainland China, almost all of which were noted to be affiliated with hospitals (147/159, 92%) (see the Materials and Methods section) (Table 1). In contrast, less than half (6/15, 40%) SGK articles from five other countries were affiliated with hospitals (Table 1 and Supplemental Data 1).

S&B screening (https://www.protocols.io/view/seek-amp-blastn-standard-operating-procedure-bjhpkj5n) flagged 144/174 (83%) SGK articles for further analysis (Fig 1). Manual verification of the identities of all nucleotide sequences in flagged articles (see the Materials and Methods section) confirmed that 75/174 (43%) SGK articles included 1–8 wrongly identified sequences/article (Table 1 and Supplemental Data 1). The 75 problematic SGK articles analysed 24 human cancer types, most frequently brain cancer, where one to nine problematic SGK articles were identified per queried gene (Table 2). Whereas 31/75 (41%) problematic SGK articles have been reported in earlier studies (12, 13), the remaining 45 SGK articles have not been previously analysed (Table 2). The 75 problematic SGK articles were published across 42 journals, where Spandidos Publications published the highest proportion (20/75, 27%) (Supplemental Data 1). Almost all (73/75, 97%) problematic SGK articles were published by authors from China, most which were affiliated with hospitals (68/73, 93%) (Table 1).

Problematic SGK articles described 115 wrongly identified sequences (Table 3), where half of these sequences (57/115) targeted a gene or genomic sequence other than the claimed target, followed by incorrect "non-targeting" reagents (44/115, 38%) (Fig 2 and Table 3). The 71 incorrect targeting sequences were claimed to interrogate 20 protein-coding genes (Table S1). Most (67/115, 58%) incorrect sequences recurred across at least two SGK articles (Fig 3 and Table S1), where the most frequent incorrect reagent was a previously described "non-targeting" shRNA that is predicted to

**Table 1.** Descriptions of the targeted corpora screened by Seek & Blastn with manual verification of nucleotide sequence reagent identities.

| | Single gene knockdown (SGK) | | miR-145 | | Cisplatin + Gemcitabine (C + G) | |
|---|---|---|---|---|---|---|
| | Corpus | Problematic | Corpus | Problematic | Corpus | Problematic |
| Number of articles (% of corpus) | 174 (100%) | 75 (43%) | 50 (100%) | 31 (62%) | 100 (100%) | 51 (50%) |
| Number of journals | 83 | 42 | 35 | 25 | 48 | 31 |
| Publication year median (range) | 2015 (2006–2019)[a] | 2015 (2010–2019) | 2017 (2009–2019) | 2017 (2009–2019) | 2017 (2008–2019) | 2017 (2009–2019) |
| Journal impact factor at publication year median (range) | 2.204 (0.098–8.459) | 1.778 (0.098–5.712) | 3.34 (0.700–9.050) | 3.23 (0.700–8.278) | 3.571 (1.099–10.391) | 3.041 (1.099–8.579) |
| Number of sequences/article median (range) | 6 (0–24) | 6 (1–24) | 11 (4–46) | 10 (4–46) | 11 (2–71) | 12 (2–70) |
| Number of incorrect sequences/article median (range) | ND | 1 (1–8) | ND | 1 (1–5) | ND | 2 (1–8) |
| Articles from China proportion (%) | 159/174 (91%) | 73/75 (97%) | 44/50 (88%) | 31/31 (100%) | 90/100 (90%) | 50/51 (98%) |
| Articles from China affiliated with hospitals proportion (%) | 147/159 (92%) | 68/73 (93%) | 40/44 (91%) | 28/31 (90%) | 82/90 (91%) | 48/50 (96%) |
| Articles from all other countries affiliated with hospitals proportion (%) | 6/15 (40%) | 0/2 (0%) | 0/6 (0%) | 0/3 (0%) | 1/10 (10%) | 0/1 (0%) |
| Articles with post-publication notices[b] proportion (%) | 20/174 (12%) | 13/75 (17%) | 1/50 (2%) | 1/31 (3%) | 1/100 (1%) | 1/51 (2%) |

[a]SGK articles were published until June 2019.
[b]Post-publication notices include retractions, expressions of concern and corrections.

**Table 2.** Cancer types studied in the Single Gene Knockdown (SGK) corpus, where each cancer type corresponds to a single article.

| Gene | Previously reported SGK articles | New SGK articles |
|---|---|---|
| ADAM8 | **Liver**[a] | Breast, Breast, Colorectal, Gastric, Liver, Lung, Pancreatic |
| ANXA1 | N/A | **Breast**, Breast, Breast, Esophageal, Leukemia, Liver, Lung, Prostate |
| EAG1 | **Liposarcoma, Osteosarcoma** | **Brain, Osteosarcoma**, Osteosarcoma, **Ovarian**, Sarcoma |
| GPR137 | **Bladder, Brain, <u>Colorectal</u>**[b]**, Pancreatic** | Brain, **Gastric, Leukemia**, Liver, Osteosarcoma, **Ovarian**, Prostate |
| ICT1 | **<u>Brain</u>** | Breast, Gastric, Leukemia, Lung, **Lymphoma, Prostate** |
| KLF8 | **Osteosarcoma** | **Bladder, Brain**, Brain, Brain, Breast, Colorectal, Colorectal, **Gastric, Gastric**, Gastric, Gastric, Liver, **Liver, Nasopharyngeal, Oral,** Ovarian, Pancreatic, **Renal** |
| MACC1 | **Ovarian** | **Bladder**, Brain, **Cervical**, Cervical, Colorectal, Colorectal, **Esophageal,** Esophageal, Gallbladder, Gastric, Liver, Liver, Lung, **Oral**, Oral, Nasopharyngeal, **Ovarian**, Ovarian, **Skin** |
| MYO6 | **<u>Brain</u>, Colorectal, Liver, Lung** | **Breast**, Gastric, Oral, **Prostate** |
| NOB1 | Brain, **<u>Breast</u>**, Colorectal, Liver, **Osteosarcoma**, Ovarian, Prostate | Laryngeal, **<u>Lung</u>, Lung**, Oral, Osteosarcoma, Renal, Thyroid, Thyroid |
| PP4R1 | **<u>Breast</u>, Liver** | **Lung** |
| PP5 | **Colorectal, Ovarian** | Bladder, **Brain, Leukemia**, Liver, Osteosarcoma, Pancreatic, Prostate |
| PPM1D | Bladder, Lung | Brain, Brain, Breast, Breast, Liver, **Pancreatic** |
| RPS15A | **Brain, Lung** | Brain, **Gastric, Leukemia**, Liver, Lung, **Osteosarcoma, Renal, Thyroid** |
| TCTN1 | **<u>Brain</u>**, Brain, **Pancreatic** | Brain, Colorectal, Gastric, Thyroid |
| TPD52L2 | **Brain, <u>Breast</u>, <u>Gastric</u>, <u>Liver</u>, <u>Oral</u>** | Brain |
| USP39 | **<u>Liver</u>, Thyroid** | **Breast, Colorectal**, Colorectal, **Gastric, Liver, Liver**, Lung, **Oral**, Osteosarcoma, Renal, Skin |
| ZFX | **Brain**, Breast | Brain, Brain, Gallbladder, Laryngeal, Leukemia, Lung, Lung, Oral, Oral, **Osteosarcoma**, Pancreatic, Prostate, **Renal** |

[a]Cancer types shown in bold correspond to problematic articles with wrongly identified nucleotide sequence(s).
[b]Underlined cancer types correspond to articles that have been retracted or assigned an expression of concern.

**Table 3. Wrongly identified nucleotide sequences summarized according to experimental technique and identity error type.**

| Corpus | Technique | "Non-targeting" yet targeting proportion (%) | "Targeting" yet non-targeting proportion (%) | Targeting wrong gene/sequence proportion (%) | Total per corpus proportion (%) |
|---|---|---|---|---|---|
| SGK (n = 115 reagents in n = 75 articles) | PCR[a] | 0/45 (0) | **7/14 (50)** | **45/57 (79)** | 52/115 (45) |
| | Gene knockdown[b] | **44/44 (100)** | **7/14 (50)** | 12/57 (21) | **63/115 (55)** |
| | Other[c] | 0/45 (0) | 0/14 (0) | 0/57 (0) | 0/115 (0) |
| | Total (Error type) | 44/44 (100) | 14/14 (100) | 57/57 (100) | 115/115 (100) |
| *miR-145* (n = 49 reagents in n = 31 articles) | PCR | 0/2 (0) | **8/9 (89)** | **33/38 (87)** | **41/49 (84)** |
| | Gene knockdown | **2/2 (100)** | 1/9 (11) | 5/38 (13) | 8/49 (16) |
| | Other | 0/2 (0) | 0/9 (0) | 0/38 (0) | 0/49 (0) |
| | Total (Error type) | 2/2 (100) | 9/9 (100) | 38/38 (100) | 49/49 (100) |
| C + G (n = 109 reagents in n = 51 articles) | PCR | 0/6 (0) | **23/24 (96)** | **73/79 (93)** | **96/109 (88)** |
| | Gene knockdown | **4/6 (67)** | 1/24 (4) | 5/79 (6) | 10/109 (9) |
| | Other | 2/6 (33) | 0/24 (0) | 1/79 (1) | 3/109 (3) |
| | Total (Error type) | 6/6 (100) | 24/24 (100) | 79/79 (100) | 109/109 (100) |
| *Gene* (n = 284 reagents in n = 128 articles) | PCR | 0/9 (0) | **35/42 (83)** | **218/233 (94)** | **253/284 (88)** |
| | Gene knockdown | **9/9 (100)** | 7/42 (17) | 15/233 (6) | 31/284 (11) |
| | Other | 0/9 (0) | 0/42 (0) | 0/233 (0) | 0/284 (0) |
| | Total (Error type) | 9/9 (100) | 42/42 (100) | 233/233 (100) | 284/284 (100) |
| *Oncology Reports* (n = 995 reagents in n = 436 articles) | PCR | 0/36 (0) | **296/335 (88)** | **573/630 (91)** | **869/995 (87)** |
| | Gene knockdown | **30/30 (100)** | 37/335 (11) | 54/630 (8) | 121/995 (12) |
| | Other | 0/36 (0) | 2/335 (1) | 3/630 (1) | 5/995 (1) |
| | Total (Error type) | 30/30 (100) | 335/335 (100) | 630/630 (100) | 995/995 (100) |
| Total (n = 1,535 reagents in n = 712 articles) | PCR | 0/89 (0) | **364/416 (87)** | **937/1,030 (90)** | **1,301/1,535 (84)** |
| | Gene knockdown | **87/89 (98)** | 50/416 (12) | 89/1,030 (9) | 226/1,535 (15) |
| | Other | 2/89 (2) | 2/416 (1) | 4/1,030 (1) | 8/1,535 (1) |
| | Total (Error type) | 89/89 (100) | 416/416 (100) | 1,030/1,030 (100) | 1,535/1,535 (100) |

[a]PCR = Human gene or genomic targeting primers for PCR, RT–PCR or methylation-specific PCR.
[b]Gene knockdown = siRNA or shRNA.
[c]Other = Claimed Ribozyme, TALEN, mimic sequences, and other oligonucleotide sequences.
Bold text indicates the most frequent error types per corpus.

target *TPD52L2* (11, 12, 13). This shRNA or highly similar variants were used as "non-targeting" controls in 41% (31/75) problematic SGK articles (Table S1).

### miR-145 corpus

Articles that focussed upon *miR-145* were identified using PubMed similarity searches of index articles (12, 13) and keyword searches of the Google Scholar database (see the Materials and Methods section). A total of 163 *miR-145* articles were then screened by S&B to flag 50 *miR-145* articles for further analysis (Fig 1). The 50 flagged *miR-145* articles were published between 2009 and 2019 across 35 journals (Table 1) and examined 18 human cancer types, where a single cancer type was analysed in each article (Supplemental Data 2). All flagged articles examined *miR-145* in combination with one to five other human gene(s) that were named in publication titles, with a minority of titles (5/50, 10%) also naming a single drug (Supplemental Data 2). Most (44/50, 88%) flagged *miR-145* articles were

published by authors from China, where most articles (40/44, 91%) were also affiliated with hospitals (Table 1 and Supplemental Data 2). The 6 *miR-145* articles from five other countries were affiliated with institutions other than hospitals (Table 1 and Supplemental Data 2).

Manual verification of S&B results revealed that most (31/50, 62%) flagged *miR-145* articles described at least one wrongly identified sequence, with a median of one (range 1–5) incorrect sequence/article (Table 1 and Supplemental Data 2). The 31 problematic *miR-145* articles were published from 2009 to 2019 across 25 journals, with the highest proportion published by Wiley (Supplemental Data 2). The 31 problematic *miR-145* articles analysed 12 human cancer types, most frequently colorectal or lung cancer, and described 49 wrongly identified sequences, most of which (38/49, 78%) targeted a different gene or target from that claimed (Fig 2 and Table 3). The 47 incorrect targeting sequences (Table 3) were claimed to interrogate 13 protein-coding and 4 non-coding RNA's (ncRNA's) (Table S1). In contrast to SGK articles, most

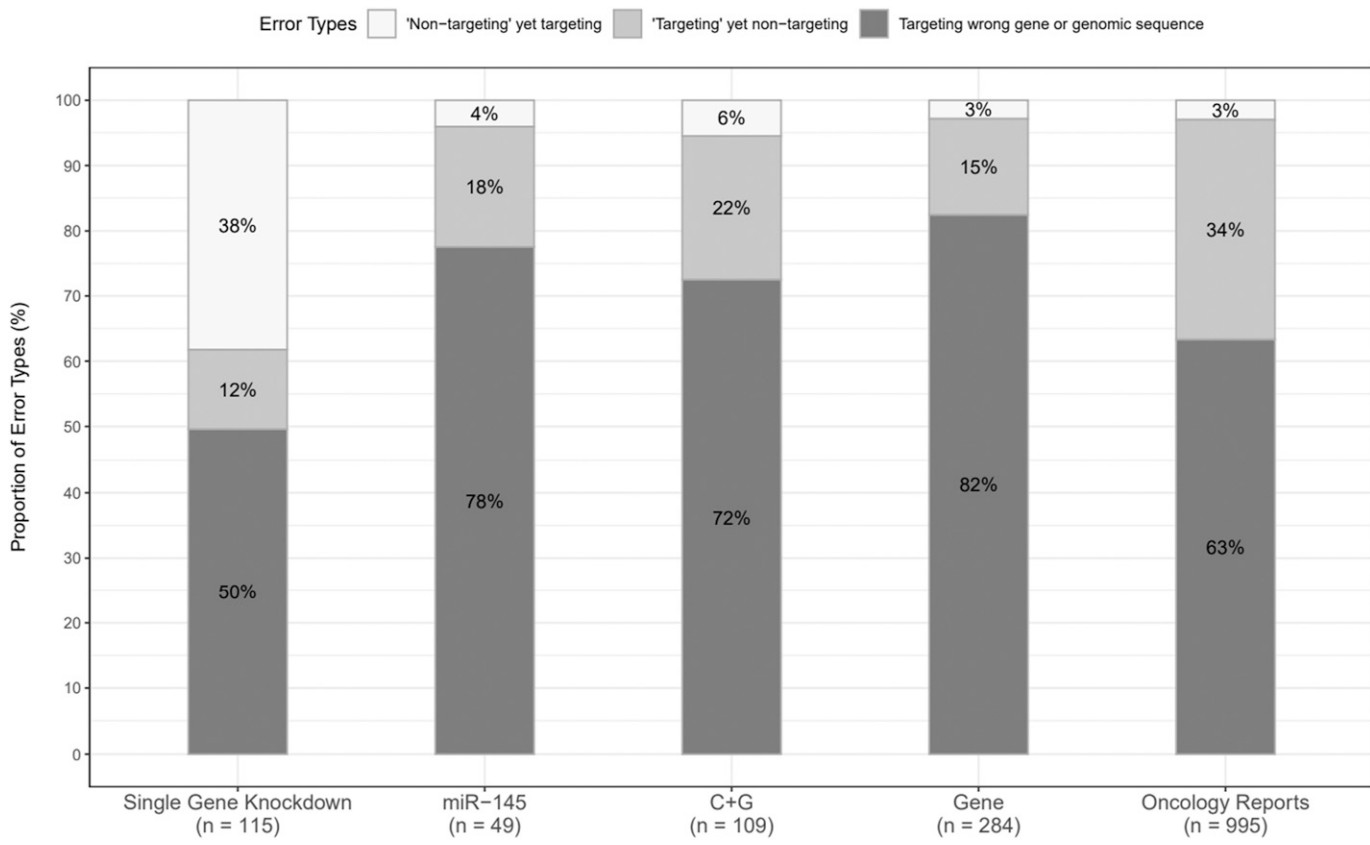

**Figure 2. Percentages of sequence identity error types in each corpus.**
Percentages of wrongly identified nucleotide sequence reagents that correspond to the three identity error types (y-axis) in each corpus (x-axis). Percentages corresponding to each error type are indicated, rounded to the nearest single digit. The numbers of incorrect sequences in each corpus are shown below the x-axis.

incorrect sequences in *miR-145* articles were used as (RT)-PCR primers (Table 3) and were identified only once within the corpus (Fig 3). All problematic *miR-145* articles were published by authors from China, where almost all articles (29/31, 94%) were affiliated with hospitals (Table 1 and Supplemental Data 2).

## Cisplatin and gemcitabine (C + G) corpus

PubMed similarity searches of index articles (12, 13) combined with keyword searches of the Google Scholar database were used to identify articles that described either cisplatin or gemcitabine treatment of human cancer cell lines and/or cancer patients (see

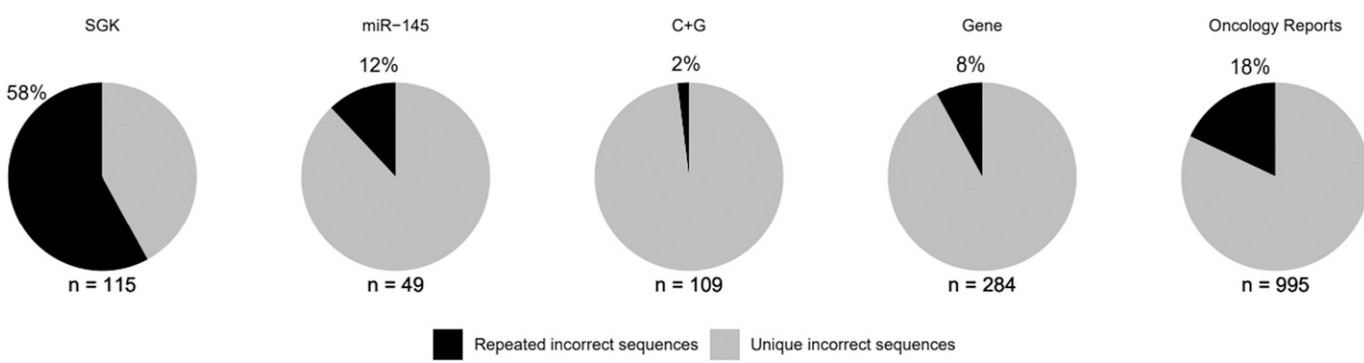

**Figure 3. Percentages of wrongly identified nucleotide sequences that were either unique or repeated within each corpus.**
Percentages of wrongly identified sequences that were identified at least twice in any single corpus (black) are shown above each image, rounded to the nearest single digit. All other wrongly identified sequences were unique in the indicated corpus (grey). Numbers of wrongly identified sequences identified in each corpus are shown below each image.

the Materials and Methods section) (Fig 1). A total of 258 articles were screened by S&B to flag 100 articles (n = 50 articles for each drug) for further analysis as a combined cisplatin + gemcitabine (C + G) corpus (Fig 1). The 100 flagged C + G articles were published between 2008 and 2019 across 48 journals (Table 1) and referred to a median of 2 (range 0–4) human genes across their titles (Supplemental Data 3). The 100 flagged C + G articles examined 13 human cancer types, where most (96/100, 96%) examined a single cancer type, typically pancreatic (35/100, 35%) or lung (22/100, 22%) cancer (Supplemental Data 3), reflecting the clinical use of cisplatin and gemcitabine (27, 28). Most (90/100, 90%) C + G articles were published by authors from China, where most (82/90, 91%) were also affiliated with hospitals (Table 1 and Supplemental Data 3). In contrast, 1 of 10 C + G articles from eight other countries was hospital-affiliated (Table 1 and Supplemental Data 3).

Approximately half (51/100, 51%) the flagged C + G articles were found to include a median of 2 (range 1–8) wrongly identified sequences/article (Table 1). The 51 problematic C + G articles were published between 2009 and 2019 across 31 journals (Table 1), where Springer Nature published the highest proportion (15/51, 29%), followed by Elsevier (12/51, 24%) (Supplemental Data 3). The 51 problematic C + G articles examined 13 human cancer types, most frequently pancreatic cancer, and described 109 wrongly identified nucleotide sequences, most of which (79/109, 72%) targeted a gene or genomic sequence other than the claimed target (Fig 2 and Tables 3 and S1). The 103 incorrect targeting sequences (Table 3) were claimed to interrogate 31 protein-coding genes and 16 ncRNA's (Table S1). As in *miR-145* articles, most incorrect sequences in problematic C + G articles represented (RT)-PCR primers (Table 3) and were identified once within the corpus (Fig 3). Almost all (50/51, 98%) problematic C + G articles were published by authors from China, where almost all (48/50, 96%) were affiliated with hospitals (Table 1 and Supplemental Data 3).

### Analysis of *Gene* and *Oncology Reports* corpora

S&B was used to screen all original articles published in *Gene* from 2007 to 2018, and all open-access articles published in *Oncology Reports* from 2014 to 2018 (Table 4). Screening 7,399 original *Gene* articles from 2007 to 2018 flagged 742 (10%) articles for further analysis (Fig 1) (see the Materials and Methods section). Manual verification of S&B outputs found that 17% (128/742) flagged articles described a median of two (range 1–36) wrongly identified sequences/article (Table 4 and Supplemental Data 4). These 128 problematic articles referred to 186 human genes (n = 146 protein-coding, n = 40 ncRNA's) across their publication titles (Supplemental Data 4). Approximately half (65/128, 51%) the problematic *Gene* articles analysed gene function in research contexts other than human cancer, most frequently by examining gene polymorphisms in patient cohorts (16/65, 25%) (Supplemental Data 4). The remaining 60 articles analysed 17 different human cancer types, most frequently lung cancer (12/60, 20%) (Supplemental Data 4). A minority of problematic *Gene* articles (7/128, 5%) referred to drugs within their titles. Manual verification of more than 5,200 sequences highlighted 284 wrongly identified sequences across the 128 problematic *Gene* articles. Almost all (275/284, 97%) incorrect sequences represented targeting reagents (Fig 2 and Table 3) for the

analysis of 92 protein-coding genes and 24 ncRNA's (Table S1). Most (261/279, 92%) incorrect sequences were described once within the *Gene* corpus (Fig 3).

As *Oncology Reports* published many more articles per year than *Gene* from 2007 to 2018, we used S&B to screen open-access *Oncology Reports* articles from 2014 to 2018 (n = 3,778 articles, 99% *Oncology Reports* articles) (Fig 1 and Table 4). Almost half (1,709/3,778, 45%) screened articles were flagged for further analysis (Fig 1), and more than one-quarter (436/1,709, 26%) of flagged articles were confirmed to describe a median of 2 (range 1–15) wrongly identified sequences/article (Table 4 and Supplemental Data 5). Almost all (432/436, 99%) problematic *Oncology Reports* articles studied gene function in human cancer, most frequently lung (54/432, 13%) or liver cancer (46/432, 11%). A subset (51/432, 12%) of problematic articles referred to 42 different drugs across their titles, most frequently cisplatin or 5-fluorouracil (Supplemental Data 5). Manual verification of more than 5,100 sequence identities confirmed 995 wrongly identified sequences (Table S1). Almost all (965/995, 97%) incorrect sequences represented targeting reagents (Fig 2 and Table 3) for the analysis of 262 protein-coding genes and 86 ncRNA's (Table S1). Most (816/965, 85%) incorrect sequences were described once across the *Oncology Reports* corpus (Fig 3).

### Geographic, institutional, and temporal distributions of problematic *Gene and Oncology Reports* articles

The 128 problematic *Gene* articles were authored by teams from 19 countries (Fig S1A) (see the Materials and Methods section). Just over half (69/128, 54%) problematic *Gene* articles were authored by teams from China (Table 4 and Figs 4 and S1B), followed by India (10/128, 8%) and Iran (9/128, 7%) (Fig S1A). Similar results were obtained for the 95 problematic *Gene* articles from 2014 to 2018, where teams from China authored 66% (63/95) articles (Fig 4). A significantly greater proportion of problematic *Gene* articles from China were affiliated with hospitals (54/69, 78%), compared with articles from other countries (5/59, 8%) (Fisher's exact test, *P* < 0.001, n = 128 articles) (Fig 4). This difference was also noted for problematic *Gene* articles from 2014 to 2018 (Fisher's exact test, *P* < 0.001, n = 95 articles) (Fig 4).

The 436 problematic *Oncology Reports* articles were authored by teams from 13 countries (Fig S2A). Most (393/436, 90%) problematic *Oncology Reports* articles were authored by teams from China (Table 4 and Fig 4), followed by much smaller proportions from South Korea (14/436, 3%) and Japan (12/436, 3%) (Fig S2A). A significantly greater proportion of problematic *Oncology Reports* articles from China were affiliated with hospitals (342/393, 87%) compared with articles from other countries (5/43, 12%) (Fisher's exact test, *P* < 0.001, n = 436 articles) (Fig 4).

We considered the distributions of problematic *Gene* and *Oncology Reports* articles according to year of publication, country of origin, and affiliated institution type (Figs 5, S1, and S2). Problematic *Gene* articles were infrequent from 2007 to 2011 (1–4 articles/year), rising to 8–38 articles/year from 2012 to 2018, where the highest number of problematic articles was identified in 2018 (Fig 5). These numbers correspond to 1.0–4.2% of all original *Gene* articles published per year from 2012 to 2018. Articles from China represented the majority of problematic *Gene* articles from 2015 to 2018

**Table 4. Summary of features of *Gene* and *Oncology Reports* journals and problematic articles.**

| Feature | *Gene* | *Oncology Reports* |
|---|---|---|
| Publication years screened by Seek & Blastn | 2007–2018 | 2014–2018 |
| Journal impact factor (range during years screened) | 2.082–2.871 | 2.301–3.041 |
| Flagged/screened articles proportion (%) | 742/7,399 (10%) | 1,709/3,778 (45%) |
| Problematic/flagged articles proportion (%) | 128/742 (17%) | 436/1,709 (26%) |
| Incorrect sequences/problematic article median (range) | 2 (1–36) | 2 (1–15) |
| Problematic articles from China proportion (%) | 69/128 (54%) | 393/436 (90%) |
| Problematic articles from all other countries proportion (%) | 59/128 (46%) | 43/436 (10%) |
| Problematic articles from China affiliated with hospitals proportion (%) | 54/69 (78%) | 342/393 (87%) |
| Problematic articles from all other countries affiliated with hospitals proportion (%) | 5/59 (9%) | 5/43 (12%) |
| Retracted or corrected problematic articles proportion (%) | 2/128 (2%) | 2/436 (0.5%) |

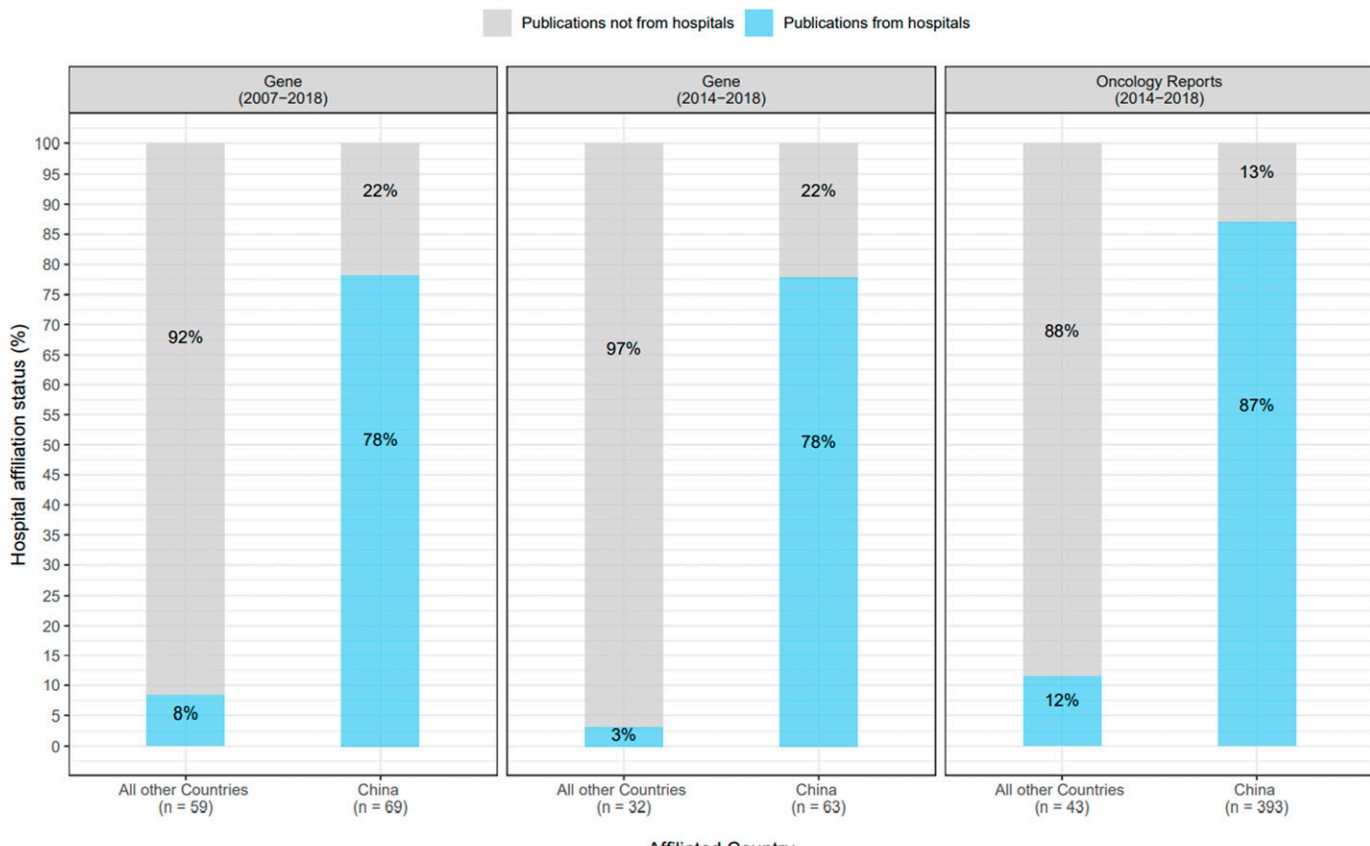

**Figure 4. Percentages of problematic *Gene* and *Oncology Reports* articles according to hospital affiliation status and country of origin.**
Percentages of problematic *Gene* and *Oncology Reports* articles according to hospital affiliation status (y-axis) from either China or all other countries (x-axis). The journal and relevant date ranges of problematic articles are shown above each panel. Problematic articles that were (not) affiliated with hospitals are shown in blue (grey), respectively. Percentages shown have been rounded to the nearest single digit. Numbers of problematic articles from China or all other countries are indicated below the x-axis. For the comparisons shown in each panel, significantly higher proportions of problematic articles from China were affiliated with hospitals versus problematic articles from other countries (Fisher's Exact test, *P* < 0.001).

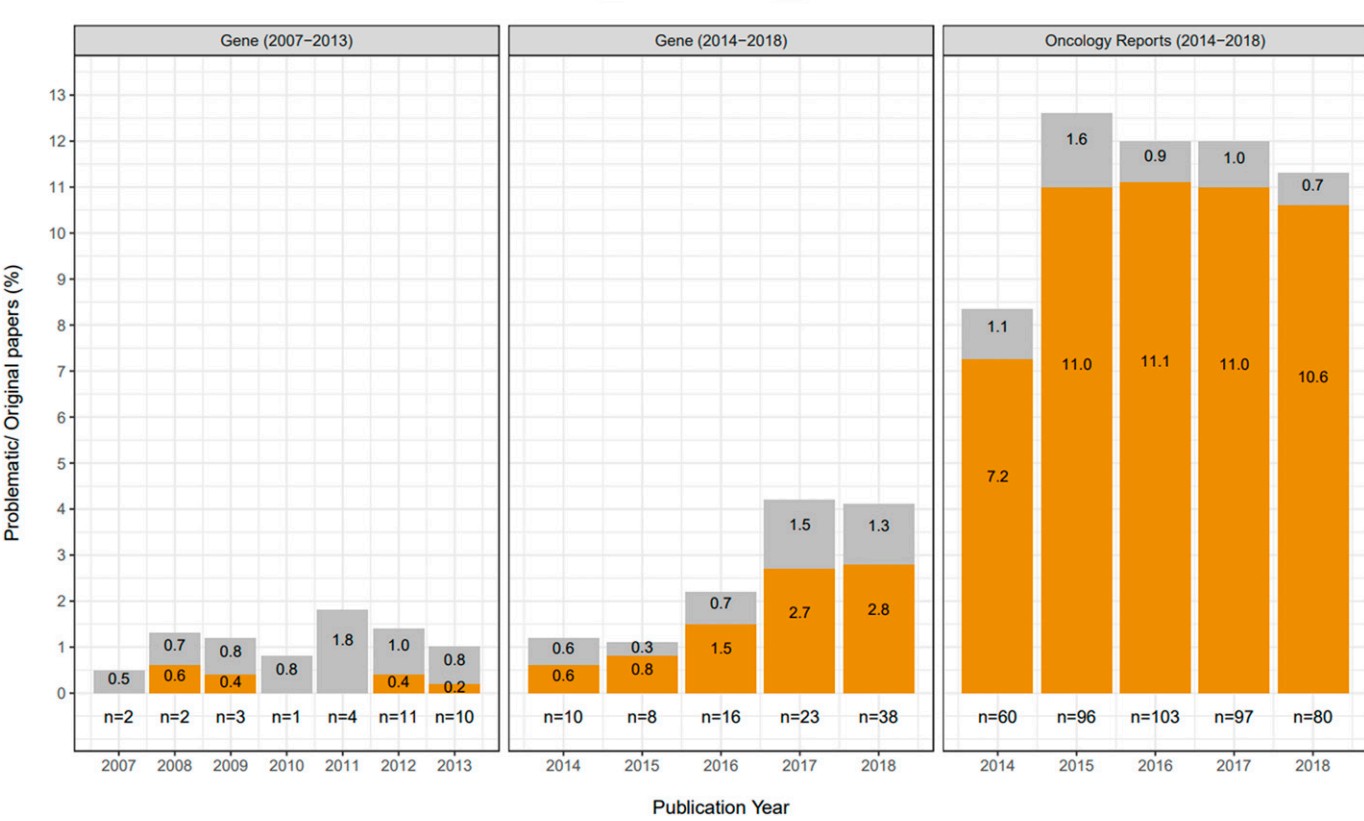

**Figure 5. Percentages and numbers of problematic *Gene* and *Oncology Reports* articles per year.**
Percentages of all *Gene* or *Oncology Reports* articles that were found to be problematic (y-axis) per publication year (x-axis). The journal and relevant publication year ranges are shown above each panel. Problematic articles from China or all other countries are shown in orange or grey, respectively. Percentages shown are rounded to one decimal place. Total numbers of problematic articles per year are shown below each graph.

(Fig 5), where most articles were also affiliated with hospitals (Fig S1B). Compared with *Gene*, *Oncology Reports* published higher numbers of problematic articles per year, corresponding to 8.3–12.6% original *Oncology Reports* articles from 2014 to 2018 (Fig 5). Across all 5 yr, most (87–93%) problematic *Oncology Reports* articles were authored by teams from China, corresponding to 11% original *Oncology Reports* articles in 2015–2017 (Fig 5), most of which were also affiliated with hospitals (Fig S2B).

### Analysis of all problematic human gene research articles

After adjusting for nine duplicate articles across the five corpora, we identified 712 problematic articles with wrongly identified sequences (Fig 1). Problematic articles were published by 31 publishers in 78 journals, most of which featured journal impact factors ≤5.0 (Supplemental Data 6). The 712 problematic articles included 1,535 wrongly identified sequences, most of which were (RT-)PCR reagents (1,301/1,535, 85%), followed by si/shRNA's (226/1,535, 15%) (Table 3).

As most incorrect reagents represented (RT-)PCR primers which are used as paired reagents, we considered the verified identities of primer pairs that were found to include at least one wrongly identified primer (Fig 6). Problematic articles frequently paired one (RT-)PCR primer that targeted the claimed gene with a primer that was predicted to target a different gene (n = 237 articles), or to be non-targeting in human (n = 118 articles) (Fig 6). Many problematic articles (n = 192) described primer pairs that were predicted to target the same incorrect gene (Fig 6). Problematic articles also combined one non-targeting primer with another that was predicted to target an incorrect gene (n = 70 articles), two non-targeting primers (n = 63 articles), and/or primers that were predicted to target two different incorrect genes (n = 42 articles) (Fig 6). Notably, 21% (276/1,301) in-correct (RT-)PCR primers were predicted to target an orthologue of the claimed gene, typically in rat or mouse (Table S1).

### Bibliometric analysis of human genes analysed in problematic articles

Almost all (1,442/1,535, 94%) incorrect sequences represented targeting reagents that were claimed to target 365 protein-coding genes and 120 ncRNA's (Table S1). The remaining 88 sequences represented incorrect "non-targeting" sequences that were instead predicted to target 35 genes, most of which (28/35, 80%) were protein-coding genes.

To count the numbers of articles in PubMed that are associated with protein-coding genes in n = 709 problematic articles (Fig 7), we used the gene2pubmed service of the National Center for

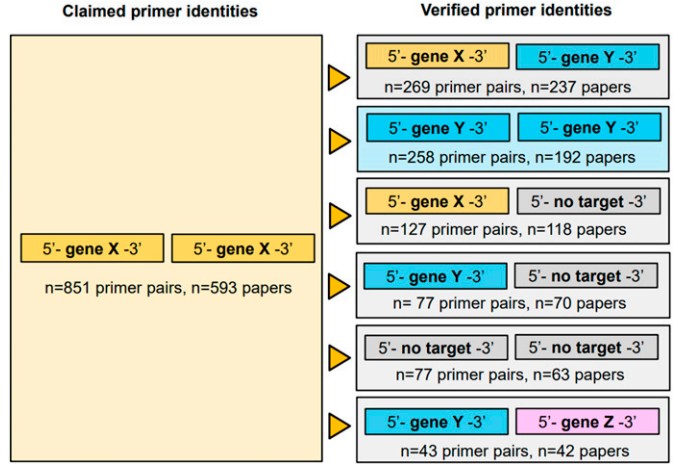

**Figure 6. Summary of (RT-)PCR primer pairings that involved at least one wrongly identified primer.**
For n = 851 primer pairs that were claimed to target particular genes/sequences (gene X) (left panel), one or both primers were predicted to be incorrect (right panel), either by targeting unrelated genes or sequences (gene Y or gene Z), or by having no predicted human target (no target). Numbers of primer pairs and affected articles are indicated below each incorrect primer pair category. Some problematic articles described more than one (category of) incorrect primer pairing. Left- or right-hand primers are not intended to indicate forward or reverse primer orientations.

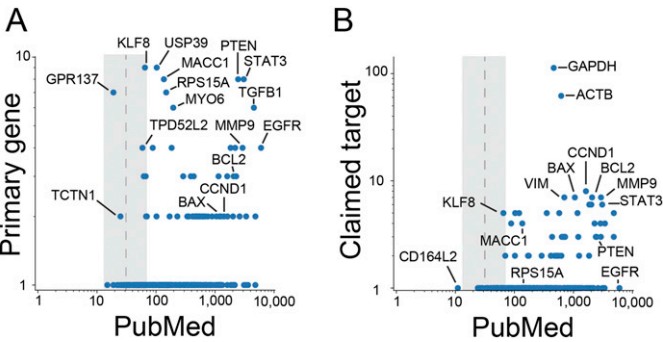

**Figure 7. Numbers of past research articles that have studied human protein-coding genes in problematic articles.**
**(A, B)** Numbers (log base 10) of problematic articles (y-axis) versus past research articles (x-axis) for (A) primary protein-coding genes in problematic articles and (B) claimed protein-coding gene targets of wrongly identified reagents. Vertical dashed lines indicate the median number of research articles for protein-coding genes, with the associated interquartile range shown in grey. Subsets of protein-coding genes are highlighted in each panel.

Biotechnology Information (31), restricting these analyses to protein-coding gene identifiers that mapped to official gene names. Primary protein-coding genes, which represented the first-listed genes in publication titles or abstracts, tended to be associated with more articles in PubMed than a randomly chosen human protein-coding gene (median publication numbers: 167 versus 31, $P < 10^{-109}$, two-sided Mann–Whitney $U$ test) (Fig S3A). Only two genes that were the primary focus of least two problematic articles (*TCTN1* and *GPR137*) (Table 2) have appeared in fewer publications in PubMed than a randomly chosen human protein-coding gene (Fig 7A). We repeated these analyses to examine the protein-coding genes that were claimed as targets by wrongly identified reagents. Again, most wrongly identified target genes have appeared in more articles than a randomly chosen protein-coding gene (median publication numbers: 238 versus 31, $P < 10^{-94}$, two-sided Mann–Whitney $U$ test) (Fig S3B). The most frequent wrongly claimed gene targets were *GAPDH* and *ACTB* (Fig 7B).

### Post-publication correction, citation, and curation of problematic articles

We considered whether any problematic articles have been the subject of post-publication notices, such as retractions, expressions of concern or corrections (11). Only 2% (11/712) of problematic articles have been retracted, where most (8/11) retraction notices did not refer to wrongly identified sequence(s), and three problematic articles have been subject to expressions of concern (Table S2). Although we excluded articles in which incorrect sequences had been subsequently corrected (see the Materials and Methods

section), we noted five corrections to problematic articles that addressed issues other than incorrect sequences (Table S2).

We then considered how problematic articles have been curated within gene knowledge bases and cited within the literature. Between 1 and 207 problematic articles were found within five gene knowledge bases that rely upon text mining (32, 33, 34, 35, 36), where knowledge bases of miR functions contained the most problematic articles (Table S3). In March 2021, the 712 problematic articles had been cited 17,183 times according to Google Scholar. Subsets of problematic C + G, *Gene* and *Oncology Reports* articles have also been cited by one or more clinical trials (Fig 8A). Given expected publication delays between pre-clinical and clinical research, we extended these data by considering the approximate potential to translate (APT) for problematic articles (37) according to publication corpus (Fig 8B). The APT metric uses the combination of concepts contained within an article to infer the probability that the article will be cited by future clinical trials or guidelines (37). The average APT for problematic articles in the five corpora ranged from 15 to 35% (Fig 8B), indicating that 15–35% of problematic articles in each corpus resemble articles that will be cited by clinical research.

## Discussion

Although there have been previous reports of wrongly identified PCR primers and gene knockdown reagents in single articles or small cohorts (11, 12, 13, 14, 15, 16, 17, 18, 19, 20 21), the present study is the first to systematically fact-check the identities of nucleotide sequences in more than 3,400 research articles. Our supported application of S&B (12, 27) (https://www.protocols.io/view/seek-amp-blastn-standard-operating-procedure-bjhpkj5n) to screen three targeted corpora and two journals identified 712 articles published across 78 journals that described more than 1,500 wrongly identified sequences. These problematic articles have received >17,000 citations, including citations by human clinical trials, where approximately one quarter of problematic articles could misinform the future development of human therapies.

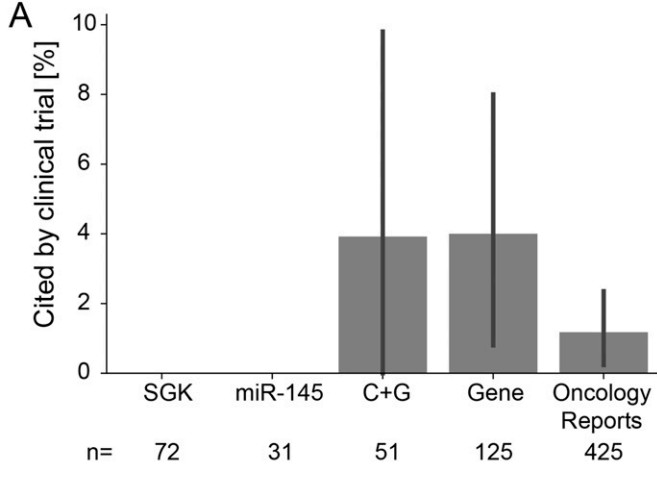

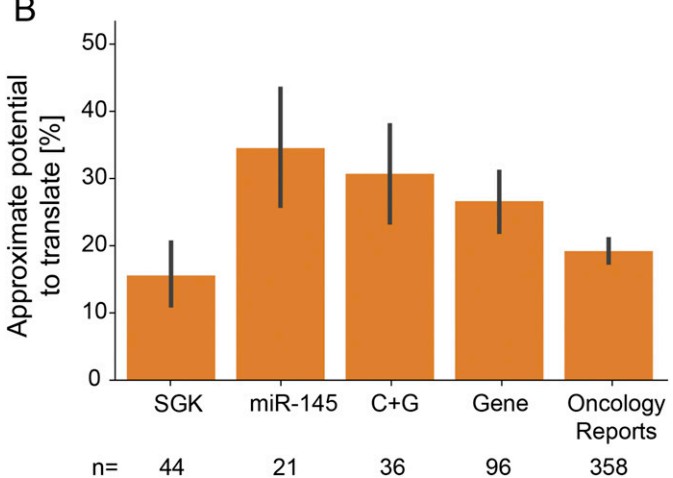

**Figure 8.   Clinical trial citations and approximate potential to translate (APT) for problematic articles.**
**(A)** Percentages of problematic articles that are cited at least once according to the NIH Open Citation Collection (y-axis), according to publication corpus (x-axis). Error bars indicate 95% confidence intervals of bootstrapped estimates of percentages. Numbers of problematic articles with at least one clinical citation are shown below the x-axis for each corpus. **(B)** Average APT for problematic articles (y-axis) according to publication corpus (x-axis). Error bars indicate bootstrapped 95% confidence intervals. Numbers of problematic articles for which the APT computed by iCite (88) are shown below the x-axis for each corpus.

Almost all (693/712, 97%) problematic articles that we identified remain uncorrected within the literature.

### Study limitations

Before discussing these results further, it is important to recognise the limitations of the methods that were used, and of particular aspects of the study design. We used the semi-automated screening tool S&B to screen original articles (12, 27), where S&B outputs for all flagged articles were manually verified to reduce false-negative and false-positive results (12, https://www.protocols.io/view/seek-amp-blastn-standard-operating-procedure-bjhpkj5n). However, as we verified the S&B outputs for 10% *Gene* articles from 2007 to 2018 and 45% *Oncology Reports* articles from 2014 to 2018, it is likely

that we did not describe all articles with wrongly identified nucleotide sequences in these journals, particularly where articles examined genes from species other than human (https://www.protocols.io/view/seek-amp-blastn-standard-operating-procedure-bjhpkj5n). We also recognise that despite taking extensive steps to verify the identities of more than 13,700 nucleotide sequences, some of the wrongly identified reagents that we have described may represent false-positives. The most challenging incorrect reagents that we encountered were claimed human targeting sequences that appeared to have no human target. As Blastn is indicated to have a very low but measurable false-negative rate (38, 39), a small fraction of what appeared to be non-targeting reagents may in fact be correct targeting reagents, as claimed. Although we therefore cannot exclude the possibility that some problematic articles have been wrongly flagged, the description of two or more wrongly identified nucleotide sequences in many problematic articles allows some protection against false-positives at the publication level.

Other limitations derive from the literature corpora that were screened. We applied S&B to targeted corpora that were identified using search terms that previously identified SGK articles from China with wrongly identified sequences (13), or index articles with wrongly identified sequences, most of which were published by authors from hospitals in China. These approaches may therefore have been likely to identify similar problematic articles in targeted corpora. We also recognise that the *miR-145* and C + G targeted corpora did not include all available articles, and so the rates of problematic articles within these corpora may not reflect the rates of problematic articles in other targeted corpora or the wider literature. Given these limitations, we applied S&B to screen original articles in *Gene* and *Oncology Reports*, without the use of search terms. Nonetheless, the selection of *Gene* and *Oncology Reports* as representative examples of journals known to have published articles with wrongly identified sequences (11, 12, 13) may have increased the likelihood of identifying problematic articles from China which were also affiliated with hospitals. Like other biomedical and genetics journals of low to moderate impact factor, articles from China constitute the majority of recent publications in both *Gene* and *Oncology Reports* (40). Moreover, as the numbers of publications from hospitals in China have risen ~50-fold from 2000 to 2020 (25), higher numbers and proportions of articles with wrongly identified sequences from hospital-based authors in China are likely to reflect these publication trends.

### Possible origins of articles with wrongly identified nucleotide sequences

As S&B screening identified more than 700 problematic articles across 78 different journals, it is important to consider the possible origins of these articles and the errors that they describe. Published errors occur in the context of both genuine and fraudulent or fabricated research, where most studies have focussed on the detection and importance of honest errors (41, 42, 43, 44). We have proposed that nucleotide sequences may be particularly prone to error, as they can be affected by different error types that are very challenging to detect by eye (8, 11, 12). As such, wrongly identified sequences are likely to represent unintended hidden errors that

persist when sequence identities are not actively checked during manuscript preparation or review (8, 11, 12). While we recognise that wrongly identified sequences in articles could also reflect some form of research sabotage, as workplace sabotage is typically directed towards known individuals (45, 46), this seems an unlikely explanation for wrongly identified sequences across hundreds of gene research articles published by many different authors.

As hidden nucleotide sequence identity errors could occur in the context of genuine and fraudulent research, some problematic articles that we have identified almost certainly represent the results of genuine research. Nonetheless, many wrongly identified sequences represented errors that seemed unlikely to occur in the context of genuine experimentation. Most incorrect reagents were (RT-)PCR primers, where most of the affected primer pairs should have failed to generate (RT-)PCR products, and yet generated results that were consistent with the experimental use of primers that targeted the claimed genes. We noted many examples of (RT-)PCR primers that were indicated to target the claimed gene in a different species, which seem unlikely errors for gene experts to commit. As previously reported (11, 12, 13), we identified articles where the non-targeting si/shRNA was verified to target the gene of interest and yet still generated the expected negative or baseline results. Such unusual reagent errors combined with implausible or impossible results could flag that some affected articles are fraudulent (8, 11, 12, 13, 24, 26).

We have previously proposed that wrongly identified nucleotide sequences in the context of repetitive and superficially justified gene research could reflect the external involvement of organisations such as paper mills (8, 13, 24, 26). Although researchers in different countries may use paper mills to meet publication targets or quotas (47), paper mills have been most widely discussed in the context of academics and medical doctors in China (25, 48, 49, 50, 51, 52, 53, 54, 55). Stringent publication requirements may represent a particular challenge for hospital doctors in China, where some hospital doctors have described limited time, training and/or opportunities to undertake research (25, 49, 50, 51, 52, 53, 54, 55). While recognising that articles with wrongly identified sequences from hospitals in China may reflect broader literature trends (25, 56), the large numbers of human gene research articles with incorrect nucleotide sequences that list hospital affiliations in China could reflect hospital doctors turning to paper mills to meet publication requirements. As most problematic articles from other countries were not affiliated with hospitals, these contrasting institutional profiles could highlight different publication pressures elsewhere. Our results combined with previous reports (25, 49, 50, 51, 52, 53, 54, 55) indicate that publication pressure upon hospital doctors in China may be exerting measurable effects upon the human gene research literature.

We have proposed that human genes represent attractive publication targets for paper mills, with many under-studied human genes (57, 58, 59) that could be targeted in different cancer or disease types that can then be distributed across different authors and journals over many years (8, 24, 26). However, whereas we had previously hypothesised that under-studied genes might be preferentially targeted by paper mills (8), in the present analysis, problematic articles rarely focussed on under-studied human protein-coding genes, and instead either focussed on or used incorrect reagents that were claimed to target protein-coding genes that had received prior attention, including highly investigated human genes such as *BCL2*, *EGFR*, *PTEN*, *STAT3*, and *CCND1*. Whereas under-studied genes may present more individual publication opportunities, articles that describe human genes with known functions and significance could carry more editorial and reader interest, increasing the likelihood of problematic manuscripts being accepted for publication and then cited by future research.

Our analyses of both targeted and journal corpora indicate that ncRNA's may provide a further layer of possibilities for the fabrication of gene-focussed articles. Although we recognise that studying genes in different diseases and analysing the functions of ncRNA's in combination with other genes are features of genuine research, our results suggest that a focus upon ncRNA's such as miR's could allow the inclusion of more topic variables within manuscripts from paper mills, such as ncRNA's and protein-coding genes that are studied across different disease types, with or without drug or natural product treatments. Examining ncRNA's in combination with other gene(s) could allow larger and more diverse publication series to be created, compared with those that focus on single genes. Furthermore, as ncRNA's possess largely numeric identifiers that may be more difficult to recognise and recall than alphanumeric protein-coding gene identifiers, any focus upon ncRNA's could contribute to large publication series being less visible within the literature. As articles that describe miR functions have also been shown to be highly cited (60), miR's and other ncRNA's could represent attractive target genes for paper mills.

## Predicted consequences of gene research articles with wrongly identified nucleotide sequences

Articles with wrongly annotated nucleotide sequence reagents could contribute to the spread of misinformation with the gene research literature, as described for incorrectly annotated genes (61, 62). Large numbers of articles that describe incorrect nucleotide sequences could encourage the incorrect selection of genes for further experimentation, possibly at the expense of more productive candidates (8). This could be exacerbated when multiple problematic articles report similar results for the analysis of the same gene (8, 13). In the present study, the identification of series of 3–20 SGK articles that universally claim that the gene target plays a causal role in 3–11 different cancer types could both individually and collectively encourage further research. Incorrect gene research articles could also lead to the overestimation of knowledge from text mining approaches, particularly given the assumed reliability of published experimental results (63), and our results demonstrate that problematic articles are already indexed within gene knowledge bases (32, 33, 34, 35, 36).

As experimental reagents, wrongly identified nucleotide sequences carry the additional risk of being wrongly used in other studies (8, 11). Although siRNA's and shRNA's are increasingly purchased from external companies as preformulated reagents, many researchers continue to order custom-made PCR primers. The most frequent incorrect reagent type that we have identified can therefore easily be reused from the literature. Experiments that either attempt to replicate published results associated with incorrect reagents and/or unknowingly reuse incorrect reagents are likely to generate unexpected results that may then remain unpublished.

As possible evidence of this, we could only identify one study that reported incorrect sequences based on the results of follow-up experiments (20) that our analyses also identified. Given that more than 17,000 citations have been accumulated by the problematic articles that we have identified, it seems inevitable that unreliable gene research articles are already wasting time and resources.

In summary, we are concerned that the number of human genes that are available for analysis, combined with research drivers that favour the continued investigation of genes of known function (57, 58, 59), are unwittingly providing an extensive source of topics around which gene research articles can be fraudulently created. Furthermore, because genuine pre-clinical gene research requires specialised expertise, time, and material resources (13), the mass production of fraudulent gene research articles could be quicker and cheaper by orders of magnitude (8). Given the number of human genes which can be studied either singly or in combination with other genes and topics such as drugs and analysed across different cancer types or other diseases, combined with unrealistic demands for research productivity (24, 47, 49, 50, 51, 52, 53, 54, 55), the publication of fraudulent gene research articles could potentially outstrip the publication of genuine gene research.

### Future directions

The number of problematic articles and incorrect reagents that we have uncovered from screening a very small fraction of the human gene research literature predicts a problem that requires urgent and co-ordinated action. Within the research community, this can take place in several ways. As the validation of nucleotide sequence identities using algorithms such as Blastn (28) represents a routine activity for teams that study individual genes, we hope that our results will encourage researchers to unfailingly check the identities of published nucleotide sequences, both in the context of their own research and during peer review. Researchers encountering wrongly identified sequences can describe these to authors, journal(s), and/or PubPeer (64) using the reporting fields that we have proposed (11). Researchers can also compare the claims of gene-focussed articles with those from high-throughput experimental studies (65) and/or predictive algorithms (63, 66). Professional societies can reinforce the importance of reagent verification through conference presentations, education programs, and journal editorials, and can advocate for tangible incentives to encourage further fact-checking of the gene research literature.

Our analysis of only a small proportion of human gene research articles, combined with the discovery that most incorrect nucleotide sequences are unique within screened literature corpora, highlights the need for further literature screening to identify other problematic gene research articles. Journals that published problematic articles in targeted corpora represent possible targets for future screening approaches, where journals with higher impact factors could be prioritized. As incorrect nucleotide sequences are unlikely to be found in all problematic gene research articles, future research should also combine analyses of nucleotide sequences with other features of concern such as manipulated or recurring experimental images and data (24, 67, 68 Preprint). Furthermore, as many research fields do not use verifiable reagents, other

methodologies are required to assess publications that describe epidemiological studies or clinical trials (69, 70).

Unfortunately, efforts from the research community alone will not solve the problem that we have described. Similarly, recent changes to researcher assessment (71, 72) will not address problematic articles that have already been published. The ability to alter the published record relies upon the engagement and co-operation of journals and publishers (11). Over the past year, growing numbers of journals have begun to recognise the issue of manuscripts and publications from paper mills (73, 74, 75, 76, 77, 78, 79, 80, 81, 82, 83), including articles that analyse genes and drug treatments in human cell lines (77, 79). Although the described efforts to screen incoming manuscripts are welcome and should be extended to all journals that publish gene research, screening incoming manuscripts must be coupled with addressing problematic articles that are already embedded in the literature (77, 80, 81, 82). These efforts could be supported by experts who could explain the significance of incorrect nucleotide sequences and/or provide training for editorial staff, particularly as the necessary researcher skills are already widely available. To overcome the protracted timeframes that can be associated with journal investigations of incorrect sequences (11), we have proposed the rapid publication of editorial notes to transparently flag articles with verifiable errors while journal and institutional investigations proceed (24).

### Summary and conclusions

Gene research relies upon correctly identified reagents to produce reliable experimental results. Wrongly identified nucleotide sequence reagents represent a threat to the continuum of genetics research, from population-based genomic sequencing to pre-clinical analyses, and the translation of these results to patients. The availability of many human genes for experimental analysis, combined with research drivers that favour the continued investigation of genes of known function (57, 58, 59), may unwittingly provide an extensive source of topics around which gene research articles can be fraudulently created (8, 24). Whereas genuine gene research requires time, expertise, and material resources, the mass production of fraudulent gene research articles by paper mills could be quicker and cheaper by orders of magnitude (8). Indeed, the possible extent of the problem of unreliable human gene research articles is indicated by the lack of overlap between the problematic articles that we have reported, and other articles of concern reported elsewhere (77, 80, 84). While publishers and journals decide how to address this problem, laboratory scientists, text miners, and clinical researchers must approach the human gene literature with a critical mindset, and carefully evaluate the merits of individual articles before acting upon their results.

# Materials and Methods

### Identification of literature corpora

#### *SGK corpus*
SGK articles were identified by combining each of 17 human gene identifiers (*ADAM8, ANXA1, EAG1, GPR137, ICT1, KLF8, MACC1, MYO6,*

*NOB1*, *PP4R1*, *PP5*, *PPM1D*, *RPS15A*, *TCTN1*, *TPD52L2*, *USP39*, and *ZFX*) with the search string "cancer AND/OR knockdown AND/OR lentivirus" (13), to search PubMed and Google Scholar databases in June 2019 using the "allintext": function for Google Scholar searches. No publication date ranges, country-specific or journal-based search terms were used to limit search results. Articles were visually inspected to confirm that articles described gene knockdown experiments that targeted one of the 17 human genes in human cancer cell lines.

### *miR-145 corpus*

The *miR-145* corpus included articles that analysed human *miR-145* function in human cell lines. Two index articles PMID 29749434 and PMID 29217166, where PMID 29217166 was verified to describe incorrect nucleotide sequence reagents (https://www.protocols.io/view/seek-amp-blastn-standard-operating-procedure-bjhpkj5n, see below), were used in PubMed similarity searches conducted in September 2019 and October 2020. Additional articles were identified through Google Scholar searches using the keywords "gene" + "miR-145" + "cancer" conducted in April 2019 and September 2020. Publication dates were limited to 2019 to broadly align with the SGK corpus. All identified articles were visually inspected to confirm the analysis of human *miR-145* function in human cell lines.

### *Cisplatin + gemcitabine (C + G) corpus*

The cisplatin + gemcitabine (C + G) corpus included articles that described either cisplatin or gemcitabine treatment of human cancer cell lines and/or biospecimens from cisplatin or gemcitabine-treated cancer patients, where most articles also reported gene research. Two index articles PMID's 30250547 and 26852750 that both described incorrect sequences (https://www.protocols.io/view/seek-amp-blastn-standard-operating-procedure-bjhpkj5n) were used in PubMed similarity searches conducted in September 2019 and October 2020. Additional articles were identified using Google Scholar searches with the search string "gene," "cancer," "cisplatin" +/– "miR" conducted in September 2019 and October 2020. A PubMed similarity search for PMID 26852750 conducted in September 2019 also identified five articles that referred to gemcitabine treatment (PMID's 18636187, 26758190, 28492560, 30117016, and 31272718). Four of these articles (PMID's 26758190, 28492560, 30117016, and 31272718) described incorrect sequences and were used as index articles for PubMed similarity searches conducted in September 2019 and October 2020. Additional publications were identified through Google Scholar searches with the query "gene," "cancer," "gemcitabine" +/– "miR" between September 2019 and October 2020. In all cases, publication dates were limited to 2019 to align with other targeted corpora. Articles were visually inspected to confirm that they studied either cisplatin or gemcitabine treatment in the context of human cancer cell lines or biospecimens, and to exclude articles from other targeted corpora.

### *Journal corpora*

*Gene* and *Oncology Reports* were selected for S&B screening as representative examples of journals that have published articles with incorrect nucleotide sequences (11, 12, 13), where *Oncology Reports* also published the highest number of problematic SGK articles (Supplemental Data 6).

*Gene* articles from January 2007 to December 2018 were retrieved using the Web of Science search criteria: PY = "2007–2018" AND SO =

"GENE" AND DT= ("Article" OR "Review"). *Oncology Reports* articles from January 2014 to December 2018 were retrieved using the Web of Science search criteria: PY = "2014–2018" AND SO = "ONCOLOGY REPORTS" AND DT= ("Article" OR "Review"). In the case of *Gene* articles, DOI's were retrieved, and PDF files were downloaded using the Elsevier Application Programming Interface with Crossref Content negotiation (http://tdmsupport.crossref.org), whereas open-access *Oncology Reports* articles were directly downloaded from www.spandidos-publications.com.

### S&B screening

SGK, *miR-145* and C + G articles were named using PMID's or journal identifiers and screened by S&B as described (12, https://www.protocols.io/view/seek-amp-blastn-standard-operating-procedure-bjhpkj5n). All SGK articles identified for the 17 selected human genes were screened by S&B. In the case of miR-145 and C + G articles, S&B screening was conducted until 50 *miR-145*, cisplatin and gemcitabine articles were flagged for further analysis, either because S&B had flagged at least one wrongly identified reagent or had failed to extract any sequences from the text. This required S&B screening of 163 *miR-145* articles and 258 C + G articles. S&B screening was conducted in 2019 and/or 2020, with all articles flagged by S&B in 2019 being rescreened by S&B in 2020.

*Gene* articles were labelled with PMID's, and batched pdf files were zipped into two compressed files according to publication dates (2007–2013 and 2014–2018). *Oncology Reports* articles were labelled by PMID's and journal identifiers. S&B screening was conducted between July and October 2019, with all articles rescreened in November 2020–February 2021. *Gene* and *Oncology Reports* articles were flagged for further analysis where S&B had either flagged at least one nucleotide sequence or had failed to extract any sequences from the text.

### Visual inspection of articles after S&B screening

Articles were visually inspected to determine the claimed genetic and/or experimental identity of each sequence. If the claimed target or experimental use of any sequence was not evident, or if a sequence was claimed to target a species other than human, the sequence was excluded from further analysis. Articles that had been subject to post-publication corrections where wrongly identified nucleotide sequences had been corrected were also excluded. We included retracted articles, to align with previous descriptions of SGK articles (11, 12, 13), and in recognition of the possibility of retracted articles continuing to be cited (85).

### Manual verification of nucleotide sequence reagent identities

Nucleotide sequence identities were manually confirmed for all sequences that were not (correctly) extracted and/or flagged as being possibly incorrect by S&B, as described (https://www.protocols.io/view/seek-amp-blastn-standard-operating-procedure-bjhpkj5n). For the *Oncology Reports* and *Gene* corpora, this involved checking at least 34% and 54% of all sequences, respectively. Further verification steps were performed for particular reagents, as follows:

(i) For reagents that were claimed to target specific gene poly-morphisms or mutant sequences and for which no sequence match could be identified by either Blastn or Blat (28, 86), manual sequence alignments were performed in Word with the query sequence in forward, forward complement, reverse and reverse complement orientations, against either the sequence corresponding to the accession number provided within the text, or to the most relevant genomic sequence found in NCBI GenBank, according to the text claim. Sequences were deline-ated using the R studio "stringr" library and accepted as targeting if the specified mutated base(s), when reverted to their original base(s) as described in NCBI dbSNP https://www.ncbi.nlm.nih.gov/snp/, allowed the reagent to target the wild-type sequence according to previously published targeting criteria (12).

(ii) If no significant matches were identified for reagents specified for the analysis of mutant or variant targets, mismatches within the nucleotide sequence were converted to the wild-type sequence, either as described in the publication or according to dbSNP and reanalysed as described (https://www.protocols.io/view/seek-amp-blastn-standard-operating-procedure-bjhpkj5n). Reagents that were indicated to target the claimed wild-type sequence were accepted as correct targeting reagents.

(iii) All flagged incorrect targeting sequences were double-checked through additional blastn searches against the database: "*Homo sapiens* (taxid:9606)," optimized for "Somewhat similar sequences (blastn)," using an expect threshold 1,000, in February 2021.

Nucleotide sequence reagents that were verified to have been wrongly identified were assigned to one of three previously de-scribed error categories (11, 12):

(i) Reagents claimed to represent targeting reagents but verified to target a human gene or target other than that claimed within the text. This error category included miR-targeting reverse RT–PCR primers with incorrect gene targeting descriptions, as supported by sequence verification (https://www.protocols.io/view/seek-amp-blastn-standard-operating-procedure-bjhpkj5n), and by having been used to analyse gene(s) other than the claimed miR and/or as a claimed universal primer according to Google Scholar searches (11, 13). Although we recognise that some of these reagents could amplify the claimed miR target as de-scribed, their descriptions as specific targeting reagents were incorrect and could lead to incorrect RT–PCR primer reuse.

(ii) Reagents claimed to target a human gene or genomic sequence but verified to be non-targeting in human. These reagents included RT–PCR primers that targeted introns or other non-transcribed regions within claimed genes.

(iii) Reagents claimed to represent non-targeting reagents in human but verified to target a human gene or genomic sequence.

### Additional publication analyses

For all articles subjected to S&B analysis, publication titles were visually inspected to identify human gene identifiers, human cancer types, and drug identifiers which were confirmed through Google searches. Human genes were categorized as either protein-coding or ncRNA's (miRs, lncRNA, or circRNA) according to Gene-Cards (https://www.genecards.org/).

Journal publishers were identified via the SCImago database (https://www.scimagojr.com/). The Journal Impact Factor corre-sponding to the (closest) publication year of each article was obtained from the Clarivate Analytics Journal Citation Reports database (40). Numbers of original articles published per year by *Gene* and *Oncology Reports* were obtained from Clarivate InCites (https://incites.clarivate.com), under Entity type = "Publication Sources," Publication Date = 2007–2018, DT = include only "Article." The country of origin of each article was assigned according to the affiliations of at least half of the listed authors. Articles were considered to be affiliated with hospital(s) if the institutional af-filiations of at least half of the listed authors were associated with one or more of the keywords: "clinic," "health cent," "hosp," "hospital," "infirmary," "sanatorium," "surgery." Articles not meeting this criterion were considered to be affiliated with institutions other than hospitals. Proportions of problematic articles (from China versus all other countries, hospitals versus other institutions) were compared using the Fisher's Exact test (SPSS statistics 27).

### Bibliometric analysis of human genes in problematic articles

Linkages of protein-coding genes to publications were obtained via gene2pubmed from the National Center for Biotechnology Information (https://ftp.ncbi.nlm.nih.gov/gene/DATA/gene2pubmed.gz) on 15 July 2021 as described (31, 58). Two-sided Mann–Whitney *U* tests were performed using SciPy (87). Post-publication notices linked with problematic articles were identified through PubMed and Google Scholar searches. PubMed ID's or other publication identifiers were used as search queries of gene knowledge bases in May 2021 (32, 33, 34, 35, 36). Publication citation counts are those reported by Google Scholar in March 2021. Problematic articles cited by clinical trials were cited by at least one pub-lication within the National Institutes of Health Open Citation collection (88), which in MedLine carried the annotation of a publication type of any "clinical trial" (without distinguishing clinical trial stage). The APT for problematic articles in each corpus was calculated as described (37) and obtained from iCite (88).

## Supplementary Information

## Acknowledgements

JA Byrne and C Labbé gratefully acknowledge funding from the US Office of Research Integrity, grant ID ORIIR180038-01-00. JA Byrne, C Labbé, and A Capes-Davis gratefully acknowledge grant funding from the National Health and Medical Research Council of Australia, Ideas grant ID APP1184263. T Stoeger gratefully acknowledges funding from the National Science Foundation, 1956338, SCISIPBIO: A data-science approach to evaluating the likelihood of fraud and error in published studies; K99AG068544, National Institutes on Aging, Integrative Multi-Scale Systems Analysis of Gene-Expression-Driven Aging Morbidity; National Institute of Allergy and Infectious Diseases,

AI135964, Successful Clinical Response In Pneumonia Therapy Systems Biology Center. The authors thank journal editorial staff for discussions and support of this study, and two anonymous peer reviewers for their insightful comments.

## Author Contributions

Y Park: data curation, formal analysis, methodology, and writing—original draft, review, and editing.

RA West: data curation, formal analysis, and writing—original draft, review, and editing.

P Pathmendra: data curation, formal analysis, and writing—review and editing.

B Favier: data curation, formal analysis, and writing—review and editing.

T Stoeger: conceptualization, data curation, formal analysis, funding acquisition, methodology, and writing—original draft, review, and editing.

A Capes-Davis: conceptualization, funding acquisition, and writing—review and editing.

G Cabanac: conceptualization, data curation, formal analysis, methodology, and writing—review and editing.

C Labbé: conceptualization, data curation, formal analysis, funding acquisition, methodology, and writing—review and editing.

JA Byrne: conceptualization, formal analysis, supervision, funding acquisition, methodology, and writing—original draft, review, and editing.

## Conflict of Interest Statement

The authors declare that they have no conflict of interest.

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
