## [Reviewer comments · Life Science Alliance]

Life Science Alliance

Identification of human gene research papers with wrongly identified nucleotide sequences

Yasunori Park, Rachael West, Pranujan Pathmendra, Bertrand Favier, Thomas Stoeger, Amanda Capes-Davis, Guillaume Cabanac, Cyril Labbé, and Jennifer Byrne

DOI: <https://doi.org/10.26508/lsa.202101203>

Corresponding author(s): Jennifer Byrne, University of Sydney

Review Timeline:

Submission Date:	2021-08-20
Editorial Decision:	2021-10-18
Revision Received:	2021-11-16
Editorial Decision:	2021-12-22
Revision Received:	2021-12-27
Accepted:	2021-12-28

Transaction Report:

October 18, 2021

Re: Life Science Alliance manuscript #LSA-2021-01203

Jennifer A Byrne
NSW Health Pathology, Faculty of Medicine and Health, The University of Sydney, NSW, Australia

Dear Dr. Byrne,

Thank you for submitting your manuscript entitled "Human gene function publications that describe wrongly identified nucleotide sequence reagents are unacceptably frequent within the genetics literature" to Life Science Alliance. The manuscript was assessed by expert reviewers, whose comments are appended to this letter. We invite you to submit a revised manuscript addressing the Reviewer comments. While revising, please pay careful attention to ensure that the tone is not inflammatory.

Thank you for this interesting contribution to Life Science Alliance. We are looking forward to receiving your revised manuscript.

Sincerely,

B. MANUSCRIPT ORGANIZATION AND FORMATTING:

Reviewer #1 (Comments to the Authors (Required)):

Park et al. present an analysis of 5 corpora using Seek & Blastn, a previously published tool. Seek & Blastn allows the authors to identify "problematic" papers in a semi-automated manner, where a paper is deemed problematic if there are typographic errors in nucleotide sequences or if sequences are incorrectly identified. The authors go on to manually verify and analyze the nature of incorrect sequences - "targeting" yet non-targeting, "non-targeting" yet targeting, and targeting wrong gene/sequence - the techniques associated with them, and how frequently wrongly identified nucleotides were repeated within corpora. The potential negative impact of problematic papers is measured via citations by clinical trials and the approximate potential to translate metric.

The authors results do suggest that wrongly identified nucleotide sequence reagents appear in the literature at a concerning frequency that is worth reporting, but I have some reservations about the manuscript. Below I divide my concerns into those about the presentation of results and analyses performed and concerns around the writing and structure of the paper in service of making the manuscript more "self-contained."

Analytical and presentation concerns

1. Three of the five corpora (SGK, miR-145, and C+G) analyzed are targeted corpora; the majority of papers in all 3 were authored by researchers affiliated with institutions in China. There are no statistics presented alongside the results in Table 1 or in the results section "Analysis of targeted publication corpora." The rationale for the genes chosen for the SGK and miR-145 corpora is also not included in the manuscript under consideration to my knowledge. Taken together, it is not clear then that there more problematic papers with authors with affiliations in China in these corpora than expected by chance or if the methodology for building targeted publication corpora was insufficiently broad, despite no geographic terms being included in the search. I recommend that the authors report the p-values for Fisher Exact tests (as was done in later sections) and the rationale for the genes chosen for the SGK and miR-145 corpora.
2. The geographic and institutional analyses (or presentation thereof) of the Gene and Oncology Reports corpora lack sufficient background information. For instance, the authors do not present what proportion of all Gene and Oncology Reports articles screened were authored by individuals with affiliations in China in Table 4. Although the authors report that a significantly greater proportion of problematic papers from China were affiliated with hospitals compared with papers from other countries, it is unclear if that result is sufficiently explained by a greater number of publications from hospitals in China compared to other institutions in China overall. I recommend that the authors present any available background information in Table 4 or include panels/facets for all papers in the corpora in Fig. 4-5 and perform statistical tests as appropriate.
3. Analyses of features indicative of manuscript templates (e.g., titles of flagged papers) could be strengthened by including comparisons to manuscripts in the corpora that were not deemed problematic. A supplemental diagram that illustrates these features, with problematic and non-problematic examples, may be helpful.

Framing and structural concerns

4. The geographic and institutional analyses of problematic papers do not seem well-justified when first encountered, as much of the context (e.g., prior work) is not introduced until the discussion of the paper. (Those analyses can also be improved in my opinion, as I detail above.)
5. Similarly, the authors state in the discussion that problematic manuscripts have features consistent with manuscript templates (pg. 24, lines 514-518). I believe that this may be supported by results in Table 2 and the analysis of titles for flagged papers in the miR-145 corpus, but a reader that is not familiar with the prior literature on manuscript templates may need more context upfront.

I recommend addressing both points by including relevant information, as well as some rationale for how corpora were built or selected, in the introduction of the manuscript.

Reviewer #2 (Comments to the Authors (Required)):

To my view, the paper submitted by Park and his collaborators deserves to be published with the highest priority. The message it conveys is of considerable importance.

Briefly, this paper demonstrates that if one takes a series of specific topics of medical interest or a couple of journals (in the fairly low IF range), one observes that a considerable number of nucleic acid sequences used to back up experiments, with important conclusions in terms of understanding topics of medical interest, the sequences do not fit with the description of the genomic objects they are supposed to tag.

However, in order to increase its likely impact, the clumsy way in which the results are presented needs to be considerably improved. In particular, perhaps because the authors were stunned by their observations, the somewhat emotional stance of the paper needs to be toned down. This means that the paper should better separate the results and the discussion, and that the introduction should be rewritten.

The main conclusion of this paper is that the nucleic acid sequences used in the analysed work do not correspond to the questions asked and the explicit experiments described. This means that statements about gene functions do not really fit the purpose of this article. The very concept of function is quite deep, difficult and far from universally shared (see e.g. C. Allen et al Nature's purposes, MIT Press 1998). This article deals with sequences, not explicitly with genes in all the cases proposed, and nucleic acid sequences can have a variety of roles, with what is currently termed "gene" coding for specific gene products, usually proteins. In this case, the function is not attributed directly to the gene (only by inference) but to its products, etc.

Nucleic acid sequences are used as probes (proxies) to identify the expression or role of a genomic object, or of an RNA for example. The issue of associated annotations is different from that explored in this article. Here too, it is known that erroneous annotations play a very negative role (see the work of Christos Ouzounis and other similar papers), and the fact that the nucleic acid sequences are wrong or erroneous has a consequence on the annotations: this should be established in the discussion section, not in the introduction, etc.

The introduction, in my opinion, should state the question posed, and outline the methods used (especially the rationale for the choice of topics explored), with a sentence or two summarising what is shown and discussed later. Then the results should present what was observed, without comment, simply highlighting the hard facts.

The discussion is very important. It should start with a summary of the main observations, and then continue with a discussion of the reasons for the choice of topics, as well as the choice of journals. The choice and quality (including drawbacks) of the methods used should also be explicitly discussed. It should also be made clear that errors are always present, but that falsification of results is particularly negative. Why is the work apparently limited to journals with a low impact factor? Recalling the work of John Ioannidis, one would expect the observations of the paper to extend to these journals, and their disastrous consequences to have proportionately even more impact.

Finally, the discussion should end with some ideas about the reasons that led to the current situation.

In the conclusion, the authors could point out that this work only explores the tip of the iceberg, and that other types of investigation (including epidemiology and clinical research) are likely to be affected by the same misinformation in ways that would be much harder to pin down.

I may be wrong, but I suspect that a more direct and explicit focus on the results as they are would have a considerable impact. This implies that the introduction should be rewritten to be incisive, rather than trying to cover all sorts of (obviously important) topics.

Dr Eric Sawey,
Executive Editor, *Life Science Alliance*

16 November 2021.

Dear Dr Sawey,

Thank-you very much for forwarding the reviewers' helpful and insightful comments on our manuscript LSA-2021-01203. We are grateful for this opportunity to reply to these comments, shown in italics below, with our responses shown in ordinary font. In some cases, we have divided longer reviewer comments into sections, to ensure that we have addressed all aspects of each comment. New text and references are shown in red in the revised manuscript, whereas previous text and references that have been reordered are shown in black. Page numbers refer to the revised manuscript, unless otherwise indicated. As described below, we have also made minor changes to the title, abstract and formatting to meet journal requirements.

While the revised version has involved rewriting and reorganizing sections of the Introduction and Discussion, no aspect of the data or results included in the submitted version have been changed.

Reviewer #1:

Park et al. present an analysis of 5 corpora using Seek & Blastn, a previously published tool. Seek & Blastn allows the authors to identify "problematic" papers in a semi-automated manner, where a paper is deemed problematic if there are typographic errors in nucleotide sequences or if sequences are incorrectly identified. The authors go on to manually verify and analyze the nature of incorrect sequences - "targeting" yet non-targeting, "non-targeting" yet targeting, and targeting wrong gene/sequence - the techniques associated with them, and how frequently wrongly identified nucleotides were repeated within corpora. The potential negative impact of problematic papers is measured via citations by clinical trials and the approximate potential to translate metric.

The authors results do suggest that wrongly identified nucleotide sequence reagents appear in the literature at a concerning frequency that is worth reporting, but I have some reservations about the manuscript. Below I divide my concerns into those about the presentation of results and analyses performed and concerns around the writing and structure of the paper in service of making the manuscript more "self-contained."

Analytical and presentation concerns

1. Three of the five corpora (SGK, miR-145, and C+G) analyzed are targeted corpora; the majority of papers in all 3 were authored by researchers affiliated with institutions in China. There are no statistics presented alongside the results in Table 1 or in the results section "Analysis of targeted publication corpora."

Response: While considered performing these analyses, we did not do so, for several reasons. As now indicated, we recognized that our selected search terms may have biased our results for targeted corpora (Introduction, page 9, 2nd paragraph, Discussion, page 22, 2nd paragraph). As such, the smaller targeted corpora included few papers from countries other than China, rendering statistical comparisons unreliable. For this reason, we reported the data only, and we have added details for problematic SGK papers which were not included in the original text (page 12). As journal screening did not rely upon search terms or specific index papers, and identified larger numbers of problematic papers, we performed statistical analyses as described.

The rationale for the genes chosen for the SGK and miR-145 corpora is also not included in the manuscript under consideration to my knowledge.

Response: This information was previously included in the Methods and Results, but in response to comments from both reviewers, we have moved this information to the Introduction (end page 8-page 9).

Taken together, it is not clear then that there more problematic papers with authors with affiliations in China in these corpora than expected by chance or if the methodology for building targeted publication corpora was insufficiently broad, despite no geographic terms being included in the search.

Response: We have now more clearly described the methodology used to build the different targeted corpora, and how the associated limitations drove our decision to screen two journals (end page 8-page 9). We have also discussed how our decision to screen two journals which were known to have published papers with incorrect sequences and how this may have influenced the types of papers that were identified (page 22).

I recommend that the authors report the p-values for Fisher Exact tests (as was done in later sections) and the rationale for the genes chosen for the SGK and miR-145 corpora.

Response: As described above, we refrained from reporting p values for the targeted corpora, given their small sizes and skewed compositions. We have also moved the rationale for gene selection from the Results to the Introduction (pages 8-9).

2. The geographic and institutional analyses (or presentation thereof) of the Gene and Oncology Reports corpora lack sufficient background information. For instance, the authors do not present what proportion of all Gene and Oncology Reports articles screened were authored by individuals with affiliations in China in Table 4.

Response: We have now provided more information to justify the selection of *Gene* and *Oncology Reports* for journal screening (Introduction, page 9), and critically discussed the selection of these journals (page 22).

The issue of which denominator to compare proportions of problematic papers is one to which we gave much thought. While we considered presenting proportions of problematic papers according to country of origin and institution type, we instead chose to report problematic papers as proportions of all original papers published in *Gene* and *Oncology Reports* (Fig 4). This represents a conservative baseline measure that can be transparently and unambiguously calculated by other studies, facilitating comparisons between the present results and those of future studies. We remain concerned that reporting proportions of papers screened from individual countries and institution types could be viewed as inflammatory, and we note the Editor in Chief's request to avoid such language in the revised manuscript. We also note that the absence of wrongly identified sequences does not denote papers as non-problematic, due to false-negative S&B results (page 21) and other possible hallmarks of problematic papers that would have required very different methodologies to assess (page 28, 2nd paragraph). Any focus on specific journal cohorts might wrongly imply that certain proportions of papers are non-problematic, which could mislead future research. We have however acknowledged that most recent *Gene* and *Oncology Reports* originate from China, according to data available through Clarivate Analytics (Discussion, page 22, ref. 41).

Although the authors report that a significantly greater proportion of problematic papers from China were affiliated with hospitals compared with papers from other countries, it is unclear if that result is sufficiently explained by a greater number of publications from hospitals in China compared to other institutions in China overall. I recommend that the authors present any available background information in Table 4 or include panels/facets for all papers in the corpora in Fig. 4-5 and perform statistical tests as appropriate.

Response: It is correct that the numbers of publications from hospitals in China have risen steeply over the past 20 years (ref. 25), and we have now indicated that the higher numbers and proportions of papers with wrongly identified sequences from hospital-based authors in China are likely to reflect these publication trends (end page 22). We have reiterated this point later in the Discussion and cited one new reference that describes papers in high impact journals from hospitals in China (ref. 57) (page 24, lines 500-501). To our best knowledge, there is no background information to inform the proportions of papers from hospitals versus other institutions in China versus other countries over relevant time periods. This would be valuable research to undertake across the preclinical and clinical research literature.

3. Analyses of features indicative of manuscript templates (e.g., titles of flagged papers) could be strengthened by including comparisons to manuscripts in the corpora that were not deemed problematic. A supplemental diagram that illustrates these features, with problematic and non-problematic examples, may be helpful.

Response: As described in our response to Q. 2, while we are confident that we have identified a set of problematic papers, as defined by the presence of wrongly identified sequence(s), we are not at all confident that all papers with correctly identified sequences are non-problematic. This is because of the possibility of false-negative Seek & Blastn results (page 21), and because papers can be problematic for many reasons beyond wrongly identified sequences (page 28, 2nd paragraph). Until gold standard sets of problematic and non-problematic papers are available for comparisons, we will not be able to test whether title analysis can help to predict whether a paper is problematic. We therefore used titles to describe the topics covered by problematic papers, as opposed to as features of problematic papers. We have retained this approach in the revised manuscript.

Framing and structural concerns

4. The geographic and institutional analyses of problematic papers do not seem well-justified when first encountered, as much of the context (e.g., prior work) is not introduced until the discussion of the paper. (Those analyses can also be improved in my opinion, as I detail above.)

Response: We have now introduced these concepts in the Introduction (pages 7-9). We have made other changes to the Discussion (please see response to Q. 2) that also improve this section (page 22).

5. Similarly, the authors state in the discussion that problematic manuscripts have features consistent with manuscript templates (pg. 24, lines 514-518). I believe that this may be supported by results in Table 2 and the analysis of titles for flagged papers in the miR-145 corpus, but a reader that is not familiar with the prior literature on manuscript templates may need more context upfront.

I recommend addressing both points by including relevant information, as well as some rationale for how corpora were built or selected, in the introduction of the manuscript.

Response: We have moved information about features of problematic gene research papers from the Discussion to the Introduction (page 7). We have removed the previous reference to the possible use of manuscript templates by paper mills, as whether or not this occurs remains unknown. As indicated above, we have added information regarding how corpora were built to the Introduction. Please see our response to Q. 3 above concerning publication title analysis.

Reviewer #2:

To my view, the paper submitted by Park and his collaborators deserves to be published with the highest priority. The message it conveys is of considerable importance.

Response: Thank-you, we appreciate this comment.

Briefly, this paper demonstrates that if one takes a series of specific topics of medical interest or a couple of journals (in the fairly low IF range), one observes that a considerable number of nucleic acid sequences used to back up experiments, with important conclusions in terms of understanding topics of medical interest, the sequences do not fit with the description of the genomic objects they

are supposed to tag.

However, in order to increase its likely impact, the clumsy way in which the results are presented needs to be considerably improved. In particular, perhaps because the authors were stunned by their observations, the somewhat emotional stance of the paper needs to be toned down. This means that the paper should better separate the results and the discussion, and that the introduction should be rewritten.

Response: We have rewritten sections of the Introduction (pages 7-9), in response to comments from both reviewers (above and below). We have transferred text from the Results to the Discussion (summary, page 21), and removed other text that previously interpreted aspects of the results. We have also reordered and rewritten sections of the Discussion, as described below.

The main conclusion of this paper is that the nucleic acid sequences used in the analysed work do not correspond to the questions asked and the explicit experiments described. This means that statements about gene functions do not really fit the purpose of this article. The very concept of function is quite deep, difficult and far from universally shared (see e.g. C. Allen et al Nature's purposes, MIT Press 1998). This article deals with sequences, not explicitly with genes in all the cases proposed, and nucleic acid sequences can have a variety of roles, with what is currently termed "gene" coding for specific gene products, usually proteins. In this case, the function is not attributed directly to the gene (only by inference) but to its products, etc.

Response: We agree, and we have removed references to “gene function” in relation to problematic papers, typically by replacing “function” with “research”, or by removing the term entirely.

Nucleic acid sequences are used as probes (proxies) to identify the expression or role of a genomic object, or of an RNA for example. The issue of associated annotations is different from that explored in this article. Here too, it is known that erroneous annotations play a very negative role (see the work of Christos Ouzounis and other similar papers), and the fact that the nucleic acid sequences are wrong or erroneous has a consequence on the annotations: this should be established in the discussion section, not in the introduction, etc.

Response: Thank-you for highlighting these papers, of which we were not previously aware. We have added this information to the Discussion (top page 26) and cited 2 new references (refs. 62 and 63).

The introduction, in my opinion, should state the question posed, and outline the methods used (especially the rationale for the choice of topics explored), with a sentence or two summarising what is shown and discussed later.

Response: We have rewritten sections of the Introduction as suggested. The questions posed are now stated on page 8, the methods and rationale for the topics explored are stated on pages 8 and 9, and brief findings are summarized on page 10.

Then the results should present what was observed, without comment, simply highlighting the hard facts.

Response: We have now removed all previous comments from the Results, and believe that this section now only presents results, without discussion.

The discussion is very important. It should start with a summary of the main observations, and then continue with a discussion of the reasons for the choice of topics, as well as the choice of journals.

Response: The main observations are summarized on page 21. The topics and journals chosen are discussed on page 22.

The choice and quality (including drawbacks) of the methods used should also be explicitly discussed.

Response: While the submitted manuscript included discussion of study limitations, we have now more explicitly discussed the choice and quality of the methods in the revised Discussion, adding new text, and reordering previous text (pages 21-22). We have now made specific reference to possible false-positive and-negative results (pages 21-22).

It should also be made clear that errors are always present, but that falsification of results is particularly negative.

Response: We now refer to errors occurring in the context of both genuine and fabricated or fraudulent research (top page 23). The Discussion also refers to negative consequences of fabricated gene research papers, due to the stringent demands of conducting genuine gene research (page 27, 2nd paragraph).

Why is the work apparently limited to journals with a low impact factor? Recalling the work of John Ioannidis, one would expect the observations of the paper to extend to these journals, and their disastrous consequences to have proportionately even more impact.

Response: Many previously identified papers with incorrect sequences were published in journals of lower impact factor which influenced our selection of journals for screening. This is now indicated by referring to “representative” journals (page 9, line 167; page 22, line 456; page 32, line 683). However, analyses of targeted corpora also identified a minority of problematic papers in journals with $IF \geq 5.0$, now mentioned in the Results (page 18) and in the revised Discussion, with the need for future research to focus on high impact factor journals (page 28). This aim forms the basis of an ongoing project that is nearing completion.

Finally, the discussion should end with some ideas about the reasons that led to the current situation.

Response: The Discussion section includes ideas about possible origins and drivers of papers with wrongly identified sequences that are now included in a separate section (pages 23-25). We believe that these ideas should be presented before the end of the Discussion, so that possible solutions can be proposed.

In the conclusion, the authors could point out that this work only explores the tip of the iceberg, and that other types of investigation (including epidemiology and clinical research) are likely to be affected by the same misinformation in ways that would be much harder to pin down.

Response: We have indicated that we have only screened a small proportion of available papers (page 28, 2nd paragraph). The revised Discussion now recognizes that not all fields involve the use of verifiable reagents and thus other methods are required to check data reliability (page 28). We have cited two new references to support this statement (refs. 70, 71).

I may be wrong, but I suspect that a more direct and explicit focus on the results as they are would have a considerable impact. This implies that the introduction should be rewritten to be incisive, rather than trying to cover all sorts of (obviously important) topics.

Response: Thank-you, we hope that the rewritten Introduction (pages 7-10) has improved the manuscript.

We have also made other formatting changes:

- Reduced the length of the manuscript title to ≤ 100 characters including spaces
- Reduced the length of the running title to ≤ 40 characters including spaces
- Reduced the number of key words to ≤ 6 key words
- Reduced the Abstract length to ≤ 175 words
- Removed the previous Author Summary and included a Summary Blurb sentence (present tense, third person) ≤ 200 characters including spaces
- Changed the previous reference bracket format to square brackets

- Reformatted and updated cited references

Thank-you again for considering this manuscript for publication in *Life Science Alliance*, and we look forward to the journal's response.

Yours sincerely,

Jennifer A. Byrne PhD

Corresponding Author

Director of Biobanking, NSW Health Pathology
Professor of Molecular Oncology, School of Medical Sciences, Faculty of Medicine and Health, The
University of Sydney, NSW, Australia

December 22, 2021

RE: Life Science Alliance Manuscript #LSA-2021-01203R

Prof. Jennifer A Byrne
University of Sydney
67-73 Missenden Road
Camperdown 2050
Australia

Dear Dr. Byrne,

Thank you for submitting your revised manuscript entitled "Identification of human gene research papers with wrongly identified nucleotide sequences". We would be happy to publish your paper in Life Science Alliance pending final revisions necessary to meet our formatting guidelines. Please address Reviewer 2's remaining comments, however, additional analyses are not expected.

A. FINAL FILES:

B. MANUSCRIPT ORGANIZATION AND FORMATTING:

****The license to publish form must be signed before your manuscript can be sent to production. A link to the electronic license to**

publish form will be sent to the corresponding author only. Please take a moment to check your funder requirements.**

Thank you for your attention to these final processing requirements. Please revise and format the manuscript and upload materials within 7 days. If you need more time due to the holidays, please let me know.

Sincerely,

Reviewer #1 (Comments to the Authors (Required)):

Thank you to the authors for the improvements to the manuscript and for the revisions to the Discussion section, in particular. I have no additional comments at this time.

Reviewer #2 (Comments to the Authors (Required)):

As discussed in my first review this type of work deserves to be known to the community, especially at a moment when there is much reluctance by part of the general public to accept the outcome of scientific studies. The work draws interesting and important conclusions but, to my view, the sample chosen by the authors introduces a bias that makes the work less impactful than what it would be if the journals and publishers were spanning a larger domain of scientific publications. In particular there should be, at some point in the article, a note on "predatory" publishers, and what this means.

Indeed, the two journals used, Gene and Oncology Reports, belong to very different modes of publishing scientific articles. Gene is currently published by Elsevier, and is a long-standing journal. On the other hand, OR is published by a very controversial publisher. I was not familiar with this publisher and did my "due diligence", which the authors should have done. Spandidos Publications, which only publishes journals with the same editor (and co-editors, from the same family), has been linked to a controversy involving its editor, has been discussed by scientists as to whether it should be included in the list of "predatory" journals, and, most importantly, has been reported in China as a problematic publisher, possibly prone to accept papers from papermills. This is made explicit in the article "What makes a journal questionable? An analysis using China's early-warning list" (doi: 10.31235/osf.io/94v5m). Although the conclusion of that study is presented in the discussion/conclusion of this article, it should have been stated from the beginning (in the introduction) as a reason to exclude Spandidos reviews from their study, as it introduces an extremely strong bias in the samples of interest. Second, the authors should have used another journal, from a reputable publisher, to develop their study. Its impact would have been much more convincing. I must say that I am somewhat surprised that the authors, having identified Spandidos Publications as tagged by the Chinese Academy of Sciences as problematic, nevertheless use it as a proof of the existence of a problem linked to publications using sequence data in China. This makes the "demonstration" circular.

An orthogonal way to perform the study would also have led to much more convincing conclusions: the authors could have used the literature directly involving some of the genes or mi-RNAs of interest and analysed the distribution of journals, with subsequent use of their software. The outcome of this experiment would have had a considerable impact.

After having read the comments of reviewer #1 I have no further comments

Dr Eric Sawey,
Executive Editor, *Life Science Alliance*

Dear Dr Sawey,

Thank-you very much for forwarding reviewer 2's comments on our revised manuscript LSA-2021-01203. We are grateful for this opportunity to reply to their comments, shown in italics below, with our responses shown in ordinary font.

As discussed in my first review this type of work deserves to be known to the community, especially at a moment when there is much reluctance by part of the general public to accept the outcome of scientific studies. The work draws interesting and important conclusions but, to my view, the sample chosen by the authors introduces a bias that makes the work less impactful than what it would be if the journals and publishers were spanning a larger domain of scientific publications. In particular there should be, at some point in the article, a note on "predatory" publishers, and what this means.

Response: We appreciate the reviewer's interest, however we refrained from discussing predatory publishers for several reasons. There is still considerable discussion as to the definition of "predatory", and some argue that this term may not be helpful, as it can be difficult to distinguish predatory and poor-quality journals/ publishers. For example, Biagioli and Lipmann (Gaming the Metrics: Misconduct and Manipulation in Academic Research, 2019, pp.1-23) advance the view that there is a quality continuum of academic journals, where predatory journals merely represent the lowest level of this continuum. Furthermore, as predatory journals tend to be poorly visible, not widely accessed or cited, and not valued by the broader community, focusing on predatory journals or publishers can also represent a distraction from the more important question of examining the reliability of published results within the indexed literature, which is the focus of our manuscript.

Indeed, the two journals used, Gene and Oncology Reports, belong to very different modes of publishing scientific articles. Gene is currently published by Elsevier, and is a long-standing journal. On the other hand, OR is published by a very controversial publisher. I was not familiar with this publisher and did my "due diligence", which the authors should have done. Spandidos Publications, which only publishes journals with the same editor (and co-editors, from the same family), has been linked to a controversy involving its editor, has been discussed by scientists as to whether it should be included in the list of "predatory" journals, and, most importantly, has been reported in China as a problematic publisher, possibly prone to accept papers from papermills. This is made explicit in the article "What makes a journal questionable? An analysis using China's early-warning list" (doi: 10.31235/osf.io/94v5m).

Response: We agree that *Gene* and *Oncology Reports* have different histories and publishers, as described (page 9), and we are aware of discussions around Spandidos Publications as a potentially problematic publisher. However, the above-mentioned China's early warning list has only been available since December 2020, and as such played no part in our 2019 decision to screen a Spandidos Publications journal, which was instead based upon our own results (page 32). *Oncology Reports* is also similar to many biomedical journals in publishing a majority of papers from China,

being indexed by PubMed and having a similar journal impact factor to *Gene* (Table 4) and other journals that published problematic papers in our study. As a cancer researcher, I am also not aware of Spandidos Publications actively soliciting for manuscript submissions, which is a key feature of predatory publishers and journals.

Although the conclusion of that study is presented in the discussion/conclusion of this article, it should have been stated from the beginning (in the introduction) as a reason to exclude Spandidos reviews from their study, as it introduces an extremely strong bias in the samples of interest. Second, the authors should have used another journal, from a reputable publisher, to develop their study. Its impact would have been much more convincing.

Response: Given the similarities between *Oncology Reports* and *Gene* (Table 4), we do not agree that screening *Oncology Reports* introduced a strong bias in our results. In support of this statement, the results of screening *Gene* and *Oncology Reports* identified many comparable results, where most problematic papers were published by author teams in China (Table 4), with a significant enrichment of papers from hospitals compared with other institutions (Table 4, Figure 4), and where problematic papers in both journals showed very similar types of incorrect reagents (Table 3). Similarities between the problematic papers that may have arisen through different models of scientific publication support our discussion of possibly broad contributions of paper mills to the gene research literature.

I must say that I am somewhat surprised that the authors, having identified Spandidos Publications as tagged by the Chinese Academy of Sciences as problematic, nevertheless use it as a proof of the existence of a problem linked to publications using sequence data in China. This makes the "demonstration" circular.

Response: Contrary to the reviewer's statement, our manuscript does not identify Spandidos Publications as having been tagged by the Chinese Academy of Sciences as problematic. This information was not available when we decided to screen *Oncology Reports* in 2019 and is therefore irrelevant to our study design.

An orthogonal way to perform the study would also have led to much more convincing conclusions: the authors could have used the literature directly involving some of the genes or mi-RNAs of interest and analysed the distribution of journals, with subsequent use of their software. The outcome of this experiment would have had a considerable impact.

Response: This approach was indeed taken to identify targeted corpora, and we provide full details of all journals that published problematic papers in each targeted corpus.

Thank-you very much again for considering this manuscript for publication in *Life Science Alliance*,

Yours sincerely,

Jennifer A. Byrne PhD

Corresponding Author

Director of Biobanking, NSW Health Pathology

Professor of Molecular Oncology, School of Medical Sciences, Faculty of Medicine and Health, The University of Sydney, NSW, Australia

December 28, 2021

RE: Life Science Alliance Manuscript #LSA-2021-01203RR

Prof. Jennifer A Byrne
University of Sydney
67-73 Missenden Road
Camperdown 2050
Australia

Dear Dr. Byrne,

Thank you for submitting your Research Article entitled "Identification of human gene research papers with wrongly identified nucleotide sequences". It is a pleasure to let you know that your manuscript is now accepted for publication in Life Science Alliance. Congratulations on this interesting work.

DISTRIBUTION OF MATERIALS:

Again, congratulations on a very nice paper. I hope you found the review process to be constructive and are pleased with how the manuscript was handled editorially. We look forward to future exciting submissions from your lab.

Sincerely,
